# How to Fine-Tune Vision Models with SGD

**Ananya Kumar**
ananya@cs.stanford.edu

**Ruoqi Shen,**
shenr3@cs.washington

**Sébastien Bubeck**
sebubeck@microsoft.com

**Suriya Gunasekar**
suriyag@microsoft.com

## Abstract

SGD and AdamW are the two most used optimizers for fine-tuning large neural networks in computer vision. When the two methods perform the same, SGD is preferable because it uses less memory (12 bytes/parameter with momentum and 8 bytes/parameter without) than AdamW (16 bytes/parameter). However, on a suite of downstream tasks, especially those with distribution shifts, we find that fine-tuning with AdamW performs substantially better than SGD on modern Vision Transformer and ConvNeXt models. We find that large gaps in performance between SGD and AdamW occur when the fine-tuning gradients in the first "embedding" layer are much larger than in the rest of the model. Our analysis suggests an easy fix that works consistently across datasets and models: freezing the embedding layer (less than 1% of the parameters) leads to SGD with or without momentum performing slightly better than AdamW while using less memory (e.g., on ViT-L, SGD uses $\sim 33\%$ less GPU memory). Our insights result in state-of-the-art accuracies on five popular distribution shift benchmarks: WILDS-FMoW, WILDS-Camelyon, BREEDS-Living-17, Waterbirds, and DomainNet.

## 1 Introduction

Fine-tuning large pretrained models on downstream tasks has become a dominant approach in deep learning (Kornblith et al., 2019; Chen et al., 2020; Zhai et al., 2020). The two most commonly used optimizers in current practice are SGD and AdamW (Kingma & Ba, 2015; Loshchilov & Hutter, 2019)[1]. While most modern vision architectures (ViTs, ConvNeXts, and variants) increasingly use AdamW for pretraining, it is still common to use SGD for fine-tuning. Part of the appeal is that SGD is more memory and compute efficient: AdamW uses $1.33\times$ and $2\times$ as much memory per parameter as SGD with and without momentum, respectively (Ginsburg et al., 2019; Dettmers et al., 2022).At the same time, in terms of fine-tuning accuracies, prior work (Dosovitskiy et al., 2021; Steiner et al., 2021; Kumar et al., 2022) report similar performance between AdamW and SGD on ImageNet like domains that are closer to pretraining data. In contrast, we reach different conclusions when fine-tuning on datasets that are far from pretraining data or have substantial distribution shifts.

We examine 7 popular models, including vision transformers (Dosovitskiy et al., 2021; Caron et al., 2021; Radford et al., 2021), ConvNeXts (Liu et al., 2022), and ResNets (Kolesnikov et al., 2020; He et al., 2016), of different sizes and pretraining modalities. When pretrained on a large corpus and then fine-tuned, these models achieve near state-of-the-art performance on downstream benchmarks. In addition to good transfer learning, we also want our fine-tuned models to handle practical distribution shifts gracefully. So we focus on 5 distribution shift datasets that have both in-distribution (ID) and out-of-distribution (OOD) evaluations: WILDS-FMoW, WILDS-Camelyon, Waterbirds, BREEDS-Living-17, DomainNet. These were selected to capture different types of data shifts (subpopulation shifts, spurious correlations, style shifts), including two real world shifts in medical imaging and satellite remote sensing from the WILDS benchmark (Koh et al., 2021).

We find that on newer models like ViTs and ConvNeXt, AdamW can significantly outperform SGD, especially OOD. Averaged across the datasets, fine-tuning a CLIP ViT-B/16 model with AdamW gets 2.1% higher accuracy ID and 8.1% higher accuracy OOD compared to SGD (Figure 1b). These gains

---

[1]By default, we use the deep learning usage of SGD as minibatch stochastic gradient descent with momentum.

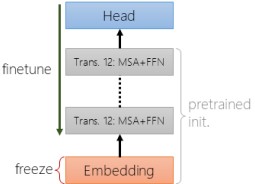

| | ID accuracy | OOD accuracy |
|---|---|---|
| SGD | 90.0% | 67.9% |
| AdamW | (+2.1%) | (+8.1%) |
| SGD (freeze-embed) | (+2.1%) | (+8.0%) |
| SGD (freeze-embed, no mom.) | (+2.2%) | **(+9.0%)** |

(a) Simplified schematic of ViT illustrating how we do freeze-embed.

(b) Performance of different fine-tuning methods on a CLIP ViT-B/16 averaged over 5 distribution shift datasets.

Figure 1: We fine-tune 7 models including ViTs, DINO, CLIP, ConvNeXt, ResNet, on 5 distribution shift datasets (Living-17, Waterbirds, DomainNet, WILDS-Camelyon, WILDS-FMoW). Fine-tuning with SGD gets lower accuracies than AdamW on modern pretrained models (Vision Transformers and ConvNeXt), especially OOD. Interestingly, a minor tweak to SGD where we freeze the first "embedding" layer ($< 1\%$ of parameters—see Figure 1a) is competitive with AdamW while using lower GPU memory. Further, dropping the momentum state in SGD gives additional gains in accuracy at even lower memory cost.

are consistent across models too—averaged across all models and datasets, AdamW gets 1.2% higher accuracy ID and 4.0% higher accuracy OOD (Tables 1-2).

A key difference between AdamW and SGD, is that AdamW normalizes the gradient update of each parameter using an estimate of their second moments. Thus, parameters with consistently high gradients will change less when using AdamW than with SGD. Towards understanding these dynamics are better, we examine the gradients at each layer of our pretrained models. We find that for the models where AdamW significantly outperforms SGD, the gradients at pretrained initialization of the first "embedding" layer are much larger than the gradients of the other layers.

To test if over-training of the embedding layer is in fact why SGD performs worse than AdamW, we consider a minor modification where we freeze the embedding layer and tune the rest of the model with SGD (Figure 1a)—we call this SGD (freeze-embed). In vision transformer models, the embedding layers are only a small fraction (around 0.7% for ViT-B/16) of the total parameters of the model, so a priori we might not expect a substantial difference in accuracies. However, surprisingly this simple freezing of the embedding layer consistently improves SGD performance across most models and datasets and achieved ID and OOD accuracies that are competitive with or better than AdamW (Figure 1b). Averaged across all datasets and models, SGD (freeze-embed) gets 76.7% accuracy OOD (vs. 72.0% for SGD and 76.0% for AdamW). The analogous AdamW (freeze-embed) gets 76.5%, which does not improve over SGD (freeze-embed), supporting that freeze-embed may be the reason AdamW outperforms SGD (it is not an independent axis of improvement). We also tried a more memory efficient variation, SGD (freeze-embed, no momentum), which drops the momentum state in SGD—interestingly this gets even slightly better OOD accuracy than the other methods (76.9%) despite using even less memory.

In terms of memory usage, our profiling (Table 4) shows that on a ViT-B/16, AdamW uses $16\%$ and $36\%$ more memory than SGD (freeze-embed) and SGD (freeze-embed, no momentum), respectively. The memory overhead of AdamW increases with the model size. On a ViT-L/14 the overheads of AdamW are $18\%$, and $49\%$, respectively.

These methods and insights, while simple, lead to state-of-the-art accuracies on all five datasets: WILDS-Camelyon, WILDS-FMoW, DomainNet, Waterbirds, and BREEDS Living-17, while being more memory efficient than AdamW.

## 2 SCOPE AND SETUP

We use the following notation: a network map $f_\theta : \mathcal{X} \to \mathcal{Y}$ is represented as a composition of layers as $f_\theta = f_{\theta_{\text{head}}}^{(\text{head})} \circ f_{\theta_L}^{(L)} \circ \ldots \circ f_{\theta_1}^{(1)} \circ f_{\theta_{\text{embed}}}^{(\text{embed})}$, where $\theta = (\theta_{\text{head}}, \theta_L, \ldots, \theta_1, \theta_{\text{embed}})$ denote all the parameters of the model. We use $f_{\theta_{\text{embed}}}^{(\text{embed})}$ and $f_{\theta_{\text{head}}}^{(\text{head})}$ to denote blocks that can conceptually be considered the "embedding" layer and the "head", respectively.

**Fine-tuning.** Consider networks that have been *pretrained* to get an initialization $\theta^{\text{pretrain}}$. We focus on *fine-tuning* on a labeled dataset $D_{\text{train}} \sim P_{\text{finetune}}$ from a new task. Concretely, given a loss

function $\ell : \mathcal{Y} \times \mathcal{Y} \to \mathbb{R}_{\geq 0}$, we minimize the training loss $L(\theta) = \frac{1}{|D_{\text{train}}|} \sum_{(x,y) \in D_{\text{train}}} \ell(f_\theta(x), y)$ using iterative optimization algorithms starting from the initialization $\theta^{\text{pretrain}}$.

We evaluate the accuracy of fine-tuned models on held-out in-distribution (ID) and out-of-distribution (OOD) test datasets. For ID evaluation, we use samples $D_{\text{test}}^{\text{id}} \sim P_{\text{finetune}}$ from the same distribution as the fine-tuning training dataset. To examine whether we have learned a robust model, we consider benchmarks that also provide OOD test examples $D_{\text{test}}^{\text{ood}} \sim P_{\text{ood}}$, which differs from the fine-tuning distribution $P_{\text{finetune}}$ in practically meaningful ways.

## 2.1 Optimizers: SGD and AdamW

The two most common optimizers for minimizing the fine-tuning loss $L(\theta)$ from pretrained initialization are SGD (with/no momentum) and AdamW. We will introduce other optimizers as needed. Compared to vanilla SGD (no momentum), which only stores the parameters and gradients as optimizer states, SGD (with momentum) stores 1 extra state per parameter (to track the first moment), while AdamW stores 2 extra states per parameter (to track the first and second moments)—see Appendix A for exact updates. This corresponds to a difference between AdamW and SGD and SGD (no momentum) of 4GB and 8GB GPU memory per 1B parameter model during training[2]. With the current scale of the models 100s of billions parameters and increasing, such memory overheads are very costly (Dettmers et al., 2022). Thus, understanding when and how we can use the cheaper SGD compared to AdamW can significantly improve training of large scale models.

## 2.2 Datasets and model architectures

**Datasets.** We choose five fine-tuning benchmarks that capture different types of data shifts (sub-population shifts, spurious correlations, style shifts), including two real world shifts.

1. **Living-17** (Santurkar et al., 2020) is a sub-population shift dataset from the BREEDS benchmark. The goal is to classify an image as one of 17 animal categories with ID and OOD data from different sub-categories. For example, in the "bear" category, the ID dataset contains black bears and sloth bears and the OOD dataset has brown bears and polar bears.
2. **Waterbirds** (Sagawa et al., 2020) is a spurious correlation dataset where the goal is to classify an image as a "waterbird" or "landbird". In the ID dataset, "water" backgrounds are typically correlated with "waterbird" labels, but are uncorrelated in the OOD dataset.
3. **DomainNet** (Peng et al., 2019) is a domain adaptation benchmark. ID contains *sketch* images, and the OOD contains *real* images of the same categories. We use the version of the dataset from Tan et al. (2020).
4. **WILDS-FMoW** (Christie et al., 2018; Koh et al., 2021) consists of remote sensing satellite images. The goal is to classify a satellite image into one of 62 geographical categories.The ID dataset contains satellite images from across the world between 2002 and 2012, and the OOD dataset contains images from Africa in 2017.
5. **WILDS-Camelyon** (Bandi et al., 2018; Koh et al., 2021) is a medical images dataset for detecting tumors in tissue slides. The ID and OOD datasets contain slides from different hospitals.

**Model Architectures.** We consider seven popular pretrained models that span different architectures (vision transformers and convolutional networks), sizes, and pretraining objectives (multi-modal, supervised, self-supervised).

(1-2) CLIP ViT-B/16 and CLIP ViT-L/14 (Radford et al., 2021): CLIP vision transformers of two sizes pretrained on a multi-modal WebImageText dataset.
(3) ViT-B/16 (Dosovitskiy et al., 2021): vision transformer pretrained on Imagenet-21k.
(4) DINO ViT-B/16 (Caron et al., 2021): self-supervised ViT pretrained on ImageNet-1K.
(5) ConvNeXt-B (Liu et al., 2022): modernized convnet pretrained on ImageNet-21k using advanced data augmentations and MixUp as in Touvron et al. (2021).

---

[2]The bottleneck for memory in older ResNe(X)ts is typically the number of activations. However, in modern large transformer training, memory requirements are of the same scale as the number of parameters. Further, techniques such as gradient accumulation and gradient checkpointing can make the activation memory small leaving the optimizer states as the main bottleneck.

(6-7) BiT ResNet-50 and BiT ResNet-101 (Kolesnikov et al., 2020): ResNetV2 models of two sizes pretrained on ImageNet-21k.

$f_{\theta_{\text{embed}}}^{(\text{embed})}$ and $f_{\theta_{\text{head}}}^{(\text{head})}$: For vision transformer models, we consider the patch-to-token embedding layer along with its layer norm if present as the *embedding layer*. For convolutional networks, the *embedding layer* refers to the "stem" block along with the first stage: the "stem" in ResNetV2 is a $7 \times 7$ convolution with stride 2 followed by a $2 \times 2$ MaxPool; while in ConvNeXt it is a non-overlapping $4 \times 4$ convolution with stride 4. For each downstream task, we replace the final layer of all the pretrained models with a randomly initialized classifier *head* $f_{\theta_{\text{head}}}^{(\text{head})}$.

## 3   SGD, ADAMW, AND LAYER GRADIENTS

Modern deep learning models increasingly use AdamW for pretraining, where it has been repeatedly shown to produce better features for downstream tasks than SGD (Dosovitskiy et al., 2021; Liu et al., 2021; 2022). For fine-tuning, on the other hand, there are no systematic studies or a definitive answer as to whether AdamW or SGD should be used (Dosovitskiy et al., 2021; Touvron et al., 2021). The ablation study in Touvron et al. (2021) even found that there is no difference in performance between AdamW and SGD when fine-tuning on ImageNet-1K. ConvNext (Liu et al., 2022) and Swin transformers (Liu et al., 2021) papers report using AdamW for fine-tuning, but they do not mention a comparison with SGD. In this work we focus on better understanding the dynamics of AdamW and SGD during the fine-tuning phase. Detailed results are discussed in Section 4. We first highlight some initial observations below.

### 3.1   ADAMW VS SGD

**AdamW outperforms SGD.**   We find that, generally speaking, fine-tuning with AdamW produces better results than with SGD, especially for more recent models like ViT variants and ConvNeXt. The gaps are more dramatic on out-of-distribution (OOD) test accuracies compared to in-distribution (ID). See Table 1 and Table 2 for the full OOD and ID results, respectively. Averaged across the 7 models and 5 datasets, AdamW gets an OOD accuracy of 76.0% (vs. 72.0% for SGD), and an ID accuracy of 91.5% (vs. 90.3% for SGD). We emphasize that this happens even though we sweep over 6 learning rates and early stop.

**AdamW $\approx$ SGD on BiT ResNets.**   Breaking down the results by model type, we find that AdamW and SGD perform comparably for older convolutional networks, namely BiT ResNet-50 and BiT ResNet-101. For example, on a BiT ResNet-101, AdamW gets an average OOD accuracy of 74.5% (vs. 74.6% for SGD). However, for newer models including vision transformers and the modernized convolutional network (ConvNeXt), AdamW gets much higher accuracies.

### 3.2   EXAMINING LAYER GRADIENTS

A key operational difference between AdamW and SGD is that AdamW divides the gradient update for each parameter by a weighted running average of its second moments. So parameters with consistently high gradients will change less when using AdamW than with SGD. This suggests examining the gradients across different components of the neural network (at pretrained initialization)—if they vary a lot, then AdamW and SGD will behave very differently.

We measure the average gradient norm at each layer, across minibatches of the training set. More formally, recall our notation for network layers as $f_\theta = f_{\theta_{\text{head}}}^{(\text{head})} \circ f_{\theta_L}^{(\text{L})} \circ \ldots \circ f_{\theta_1}^{(1)} \circ f_{\theta_{\text{embed}}}^{(\text{embed})}$. Given a minibatch $B = \{(x_1, y_1), \ldots, (x_b, y_b)\}$ of random samples from the training data, the stochastic gradient of layer $\ell \in \{\text{embed}, 1, 2, \ldots, L, \text{head}\}$ is given by: $g_\ell(B) = \frac{1}{|B|} \sum_{i=1}^{|B|} \nabla_{\theta_\ell} l(f_{\theta^{\text{pretrain}}}(x_i), y_i)$. The norm of the stochastic gradient, $\|g_\ell(B)\|_2$, roughly measures how much layer $\ell$ will change after one step of SGD from pretrained initialization. We use the average gradient norm across all minibatches $B_1, \ldots, B_m$ in the training set $D_{\text{train}}$ as a measure of movement in the first SGD step.

$$G_\ell^{\text{init}} = \frac{1}{m} \sum_{t=1}^{m} \|g_\ell(B_t)\|_2, \text{ where } \ell \in \{\text{embed}, 1 - L, \text{head}\}. \tag{3.1}$$

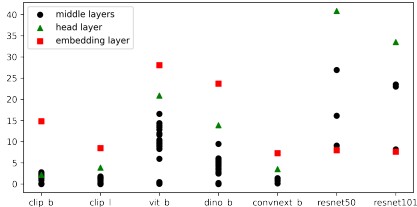

Figure 2: We visualize the layer-wise gradient norms our models on DomainNet at the pretrained initialization. For each layer, we plot the average minibatch gradient norm $G_\ell^{\text{init}}$ as computed in equation 3.1. We highlight two special layers: gradient norms of the "embedding" layer parameters $G_{\text{embed}}^{\text{init}}$ are shown as **red-squares**, and those of the classifier "head"'s $G_{\text{head}}^{\text{init}}$ are show as **green-triangles**. The middle layer gradient norms are shown as **black-circles**. For transformer models, we have separate (black) points for the MLP and the attention layers.

Note that we measure all the gradients at the pretrained initialization $\theta^{\text{pretrain}}$—before performing any gradient updates. In Figure 2, we plot the layer-wise gradient norms $G_\ell^{\text{init}}$ on DomainNet. Similar plots for other datasets and with alternative normalization are provided in Appendix F.

**First-layer gradient is an outlier.** Apriori, we expect the gradients of the classifier "head" (green-triangles) to be large as they are randomly initialized while the rest of the model is pretrained (Kumar et al., 2022). Interestingly, we see in Figure 2 (see also Appendix F) that for the recent models (ViTs and ConvNeXt), gradient norms of the "embedding" layers (red-squares) stand out as outliers with much larger gradients than the other layers. These are also the models where see big gaps between AdamW and SGD performance in Tables 1-2.

This suggests that the embedding layer plays a distinctive role in newer models. Since SGD uses the same learning rate for all layers, we will end up make substantially larger updates to the embedding layer compared to other layers—leading to either over-tuning the embedding layer or under-tuning the remaining layers. Over-training of the embedding layer is undesirable given the common wisdom that lower layers ought to be tuned less as they learn more transferable features (Kumar et al., 2022). AdamW on the other hand adaptively normalizes the movement of each parameter. On the other hand, computationally, SGD is preferable over AdamW due to lower memory footprint.

How can we avoid the above issues with SGD? One possibility is to use different learning rates for different layers, but this might end up requiring extensive hyperparameter tuning. Another option is to use other low-memory footprint optimizers with layerwise normalization techniques like LARS (You et al., 2017a) and LAMB (You et al., 2020). In our initial experiments (see Table 5), while these methods improved over SGD, they did not close the gap with AdamW. Instead, we found a much simpler modification to SGD that consistently leads to accuracies competitive with or better than AdamW while using lower memory (Section 3.4).

### 3.3 WHY FIRST-LAYER GRADIENTS ARE HIGHER?

Why are the first-layer gradients higher for modern vision models such as Vision Transformers and ConvNeXt? There are two plausible hypotheses: (a) architectural differences from ResNet models, or (b) optimization differences between *pretraining* and fine-tuning—BiT ResNet models were pretrained with SGD while the Vision Transformer and ConvNeXt models need to be pretrained with AdamW (Steiner et al., 2021). In Appendix E we run controlled experiments and find that the *optimizer mismatch* appears to be the key factor. To control for architecture, we use the same ResNet architecture and pretrain using either (1) SGD or (2) AdamW. For the SGD pretrained ResNet the first layer has smaller gradients than the other layers, while for the AdamW pretrained ResNet the first layer has higher gradients than the other layers (Figure 3). In line with our previous results, fine-tuning an AdamW *pretrained* model with SGD leads to worse OOD accuracy (Table 13), but fine-tuning an SGD *pretrained* model with SGD performs well. On the other hand, architectural changes lead to smaller changes.

### 3.4 FREEZE-EMBEDDING

While the presence of large embedding layer gradients in Figure 2 correlates with the observed performance gaps between AdamW and SGD, it is not clear that this observation or its potential

issues discussed above are definitive *causes* for SGD performing worse than AdamW. To further test the intuition, in the next section, we consider a "freeze-embed" variation of fine-tuning, where we simply freeze the embedding layer to its pretrained initialization, and then fine-tune the rest of the model as usual. The embedding layer only consists of a small fraction of the model parameters (e.g., 0.7% in a base vision transformer), so apriori freezing the embedding layer is a very tiny tweak—and we would not expect a large change in accuracy. However, if the hypothesis above holds merit, then we would expect the modification to aid SGD but not AdamW.

## 4  DETAILED EXPERIMENTS ON FREEZE-EMBEDDING

We consider a simple variation of SGD fine-tuning where we freeze the embedding layer and only perform SGD updates on the rest of the network—we call this method "SGD (freeze-embed)". For further memory gains, we also consider SGD without momentum "SGD (freeze-embed, no momentum)" In this section we discuss the results of fine-tuning our 7 models on 5 distribution shift benchmarks mentioned in Section 2. We use the implementations of SGD and AdamW in PyTorch. For each method, we train for the same number of epochs using a cosine learning rate schedule, and sweep over 6 starting learning rates (ensuring that the optimal learning rate is in the middle of the sweep). For all datasets we follow prior work (Kumar et al., 2022) and pick the best learning rate and early stop based on the ID validation accuracy. See Appendix A for additional details.

Table 1 and Table 2 show the OOD and ID accuracies, respectively, of the four methods across models and datasets. For each model and dataset, we also highlight the relative gain/loss of AdamW and SGD (freeze-embed) from regular SGD in green/red. We discuss the main observations below.

1. **AdamW outperforms SGD.** We see that AdamW largely outperforms SGD, often by substantial margins on ViT and ConvNeXt models. Particularly in OOD evaluation, the gaps are remarkable. In the few instances where SGD performs better the gaps are much smaller. The differences between SGD and AdamW are generally more modest for older ResNet models.

2. **SGD (freeze-embed) as well as SGD (freeze-embed, no momentum) are competitive with or better than AdamW.** For each individual model, averaged across the datasets SGD (freeze-embed) variants are consistently the best or tied-best method on OOD accuracy, and only minimally worse than AdamW on ID accuracy (right columns of Table 1-2). Averaged across all the models and datasets, SGD (freeze-embed) performs the best OOD, getting an average OOD accuracy of 76.7% (vs. 71.9% for SGD and 76.0% for AdamW). On ID data, SGD (Freeze-embed) closes 85% of the gap between SGD and AdamW, getting an average accuracy of 91.3% (vs. 90.2% for SGD and 91.5% for AdamW). SGD (freeze-embed, no momentum) gets the highest average OOD accuracy of 76.9% (vs. 71.9% for SGD and 76.0% for AdamW) and competitive ID accuracy of 91.2% (vs. 90.2% for SGD and 91.5% for AdamW), while saving additional memory.

3. **Larger CLIP model gains more from SGD (freeze-embed) and AdamW.** On a CLIP-ViT-B/16 we see that SGD (freeze-embed) and AdamW get about $8\%$ higher OOD accuracy than SGD. Upon scaling to a larger CLIP-ViT-L/14, which is also our best model for OOD performance, we see even higher gains. On CLIP-ViT-L/14, SGD (freeze-embed) gets a 14.3% higher OOD accuracy than SGD, and a 0.7% higher OOD accuracy than AdamW. This suggests that our findings might be even more relevant for larger models.

4. **AdamW is *red* when SGD (freeze-embed) is *red*.** Across all our models and datasets, we see that in instances where SGD (freeze-embed) is worse than SGD (i.e., highlighted as red), AdamW is also worse than SGD. This is not always the case the other way around. For example, on ViT B-/16 fine-tuned on DomainNet and Camelyon, OOD performance of AdamW is worse than SGD, but SGD (freeze-embed) is competitive to the best of the two. At the same time, on Living-17 OOD evaluation, multiple models have both AdamW and SGD (freeze-embed) perform significantly worse than SGD.

5. **SGD performs well when fine-tuning data is closer to pretraining.** Breaking down the results by datasets, the main trend we see is that with models that were pretrained on ImageNet-21k (all models here except CLIP) and fine-tuned on Living-17, the performance of SGD is typically higher than AdamW and SGD (freeze-embed). Images in Living-17 are derived from ImageNet-1K and hence arguably, in this case the fine-tuning distribution is closest to pretraining. This suggests that SGD may work better when fine-tuning and pretraining data are similar—we leave a more thorough analysis to future work.

| | | Living-17 | Waterbirds | DomainNet | FMoW | Camelyon | Avg. |
|---|---|---|---|---|---|---|---|
| CLIP ViT-B/16 | SGD | 80.0 | 62.5 | 72.8 | 37.3 | 86.8 | 67.9 |
| CLIP ViT-B/16 | AdamW | 82.8 (+2.8) | 71.9 (+9.4) | **89.2** (+16.4) | **40.7** (+3.4) | **95.7** (+8.9) | 76.0 (+8.1) |
| CLIP ViT-B/16 | SGD (freeze-embed) | **83.2** (+3.2) | 73.7 (+11.2) | 88.2 (+15.4) | 40.2 (+2.9) | 94.3 (+7.5) | 75.9 (+8.0) |
| CLIP ViT-B/16 | SGD (freeze-embed, no momentum) | 83.1 (+3.1) | **80.4** (+17.9) | 89.0 (+16.2) | 38.8 (+1.5) | 93.3 (+6.5) | **76.9** (+9.0) |
| CLIP ViT-L/14 | SGD | 84.2 | 65.0 | 60.8 | 41.0 | 83.2 | 66.8 |
| CLIP ViT-L/14 | AdamW | 88.0 (+3.8) | 85.2 (+20.2) | **93.8** (+33.0) | 48.3 (+7.3) | 95.9 (+12.7) | 82.2 (+15.4) |
| CLIP ViT-L/14 | SGD (freeze-embed) | **90.5** (+6.3) | 84.7 (+19.7) | 93.1 (+32.3) | **49.9** (+8.9) | 96.5 (+13.3) | **83.0** (+16.2) |
| CLIP ViT-L/14 | SGD (freeze-embed, no momentum) | 89.2 (+5.0) | **86.8** (+21.8) | 93.7 (+32.9) | 46.8 (+5.8) | **96.7** (+13.5) | 82.6 (+15.8) |
| Sup ViT-B/16 | SGD | **89.5** | 77.4 | **86.3** | 33.5 | 92.6 | 75.8 |
| Sup ViT-B/16 | AdamW | 88.3 (-1.2) | 81.6 (+4.2) | 84.4 (-1.9) | **35.9** (+2.4) | 87.9 (-4.7) | 75.6 (-0.2) |
| Sup ViT-B/16 | SGD (freeze-embed) | 88.0 (-1.5) | **82.4** (+5.0) | **86.3** (+0.0) | 34.4 (+0.9) | **93.7** (+1.1) | **77.0** (+1.2) |
| Sup ViT-B/16 | SGD (freeze-embed, no momentum) | 88.4 (-1.1) | 76.6 (-0.8) | 86.1 (-0.2) | 34.6 (+1.1) | 87.5 (-5.1) | 74.6 (-1.2) |
| DINO ViT-B/16 | SGD | 88.2 | 56.1 | 76.0 | 33.6 | 86.9 | 68.2 |
| DINO ViT-B/16 | AdamW | 87.4 (-0.8) | 61.2 (+5.1) | 77.4 (+1.4) | 35.8 (+2.2) | 91.9 (+5.0) | 70.7 (+2.5) |
| DINO ViT-B/16 | SGD (freeze-embed) | 86.7 (-1.5) | 67.9 (+11.8) | 78.4 (+2.4) | **35.9** (+2.3) | 90.6 (+3.7) | 71.9 (+3.7) |
| DINO ViT-B/16 | SGD (freeze-embed, no momentum) | **88.2** (+0.0) | **68.5** (+12.4) | **79.9** (+3.9) | 33.4 (-0.2) | **93.9** (+7.0) | **72.8** (+4.6) |
| ConvNext-Base | SGD | **94.0** | 80.2 | 89.8 | **39.7** | 83.0 | 77.3 |
| ConvNext-Base | AdamW | 90.3 (-3.7) | **89.8** (+9.6) | 89.5 (-0.3) | 38.4 (-1.3) | 89.5 (+6.5) | 79.5 (+2.2) |
| ConvNext-Base | SGD (freeze-embed) | 92.6 (-1.4) | 86.9 (+6.7) | 91.2 (+1.4) | 38.2 (-1.5) | 88.1 (+5.1) | 79.4 (+2.1) |
| ConvNext-Base | SGD (freeze-embed, no momentum) | 93.3 (-0.7) | 83.6 (+3.4) | **91.5** (+1.7) | 38.1 (-1.6) | **96.0** (+13.0) | **80.5** (+3.2) |
| BiT ResNet-50 | SGD | 84.3 | **76.5** | 80.0 | 34.1 | 90.4 | 73.1 |
| BiT ResNet-50 | AdamW | 83.1 (-1.2) | 74.8 (-1.7) | **84.0** (+4.0) | 33.8 (-0.3) | 92.4 (+2.0) | 73.6 (+0.5) |
| BiT ResNet-50 | SGD (freeze-embed) | **84.1** (-0.2) | 75.5 (-1.0) | 82.3 (+2.3) | **35.0** (+0.9) | 95.2 (+4.8) | 74.4 (+1.3) |
| BiT ResNet-50 | SGD (freeze-embed, no momentum) | 83.8 (-0.5) | 76.0 (-0.5) | 83.8 (+3.8) | 33.6 (-0.5) | **95.6** (+5.2) | **74.5** (+1.4) |
| BiT ResNet-101 | SGD | 82.8 | 76.9 | **86.2** | 38.0 | 89.3 | 74.6 |
| BiT ResNet-101 | AdamW | 82.9 (+0.1) | **79.4** (+2.5) | 83.5 (-2.7) | 37.0 (-1.0) | 89.7 (+0.4) | 74.5 (-0.1) |
| BiT ResNet-101 | SGD (freeze-embed) | **83.1** (+0.3) | 77.3 (+0.4) | 86.0 (-0.2) | 36.0 (-2.0) | 95.5 (+6.2) | 75.6 (+1.0) |
| BiT ResNet-101 | SGD (freeze-embed, no momentum) | 82.9 (+0.1) | 79.1 (+2.2) | 86.0 (-0.2) | 36.4 (-1.6) | **95.9** (+6.6) | **76.1** (+1.5) |

Table 1: **Out-of-distribution (OOD)** accuracies including SGD (freeze-embed) without momentum.

| | | Living-17 | Waterbirds | DomainNet | FMoW | Camelyon | Avg. |
|---|---|---|---|---|---|---|---|
| CLIP ViT-B/16 | SGD | 97.8 | 97.2 | 88.8 | 67.0 | **99.4** | 90.0 |
| CLIP ViT-B/16 | AdamW | **98.1** (+0.3) | **97.7** (+0.5) | 95.0 (+6.2) | **70.1** (+3.1) | **99.5** (+0.1) | **92.1** (+2.1) |
| CLIP ViT-B/16 | SGD (freeze-embed) | **98.2** (+0.4) | 97.8 (+0.6) | 94.9 (+6.1) | 70.0 (+3.0) | **99.5** (+0.1) | **92.1** (+2.1) |
| CLIP ViT-B/16 | SGD (freeze-embed, no momentum) | **98.2** (+0.4) | 97.9 (+0.7) | **95.2** (+6.4) | **70.1** (+3.1) | **99.5** (+0.1) | **92.2** (+2.2) |
| CLIP ViT-L/14 | SGD | 98.4 | 97.3 | 84.3 | 69.0 | 99.4 | 89.7 |
| CLIP ViT-L/14 | AdamW | **98.9** (+0.5) | 98.8 (+1.5) | 96.9 (+12.6) | **74.5** (+5.5) | **99.6** (+0.2) | **93.7** (+4.0) |
| CLIP ViT-L/14 | SGD (freeze-embed) | 98.7 (+0.3) | **98.9** (+1.6) | 97.1 (+12.8) | **74.5** (+5.5) | **99.6** (+0.2) | **93.7** (+4.0) |
| CLIP ViT-L/14 | SGD (freeze-embed, no momentum) | 98.8 (+0.4) | 98.7 (+1.4) | **97.3** (+13.0) | 74.3 (+5.3) | 99.5 (+0.1) | **93.7** (+4.0) |
| Sup ViT-B/16 | SGD | **98.6** | 99.1 | **91.7** | 64.1 | 99.4 | 90.6 |
| Sup ViT-B/16 | AdamW | **98.7** (+0.1) | 99.0 (-0.1) | **91.7** (-0.0) | **66.4** (+2.3) | 99.5 (+0.1) | **91.1** (+0.5) |
| Sup ViT-B/16 | SGD (freeze-embed) | **98.7** (+0.1) | **99.2** (+0.1) | 91.5 (-0.2) | 65.0 (+0.9) | **99.6** (+0.2) | 90.8 (+0.2) |
| Sup ViT-B/16 | SGD (freeze-embed, no momentum) | 98.5 (-0.1) | 98.9 (-0.2) | 90.6 (-1.1) | 65.7 (+1.6) | 99.5 (+0.1) | 90.6 (+0.0) |
| DINO ViT-B/16 | SGD | **98.4** | 97.0 | 88.2 | 62.4 | 99.4 | 89.1 |
| DINO ViT-B/16 | AdamW | **98.5** (+0.1) | **97.9** (+0.9) | **89.4** (+1.2) | **66.0** (+3.6) | **99.6** (+0.2) | **90.3** (+1.2) |
| DINO ViT-B/16 | SGD (freeze-embed) | **98.4** (+0.0) | 97.5 (+0.5) | 89.0 (+0.8) | 63.5 (+1.1) | 99.5 (+0.1) | 89.6 (+0.5) |
| DINO ViT-B/16 | SGD (freeze-embed, no momentum) | **98.5** (+0.1) | 97.5 (+0.5) | 89.2 (+1.0) | 63.8 (+1.4) | 99.5 (+0.1) | 89.7 (+0.6) |
| ConvNext-Base | SGD | **98.7** | 99.0 | 94.8 | 66.3 | 99.4 | 91.6 |
| ConvNext-Base | AdamW | 98.6 (-0.1) | **99.5** (+0.5) | 94.5 (-0.3) | **68.8** (+2.5) | **99.7** (+0.3) | **92.2** (+0.6) |
| ConvNext-Base | SGD (freeze-embed) | 98.6 (-0.1) | 99.4 (+0.4) | **95.1** (+0.3) | 67.4 (+1.1) | 99.5 (+0.1) | 92.0 (+0.4) |
| ConvNext-Base | SGD (freeze-embed, no momentum) | **98.8** (+0.1) | 99.2 (+0.2) | 95.0 (+0.2) | 66.0 (-0.3) | 99.4 (-0.0) | 91.7 (+0.1) |
| BiT ResNet-50 | SGD | 97.4 | 98.4 | **89.3** | 64.6 | 99.5 | **89.8** |
| BiT ResNet-50 | AdamW | 97.2 (-0.2) | **98.5** (+0.1) | 89.2 (-0.1) | **65.1** (+0.5) | **99.5** (+0.0) | **89.9** (+0.1) |
| BiT ResNet-50 | SGD (freeze-embed) | **97.6** (+0.2) | **98.5** (+0.1) | 89.2 (-0.1) | 64.8 (+0.2) | **99.5** (+0.0) | **89.9** (+0.1) |
| BiT ResNet-50 | SGD (freeze-embed, no momentum) | 97.5 (+0.1) | 98.4 (-0.0) | 89.2 (-0.1) | 63.6 (-1.0) | 99.4 (-0.1) | 89.6 (-0.2) |
| BiT ResNet-101 | SGD | 98.3 | 98.9 | **92.0** | 66.0 | 99.4 | **90.9** |
| BiT ResNet-101 | AdamW | **98.4** (+0.1) | 98.6 (-0.3) | 91.1 (-0.9) | **67.0** (+1.0) | **99.6** (+0.2) | **90.9** (-0.0) |
| BiT ResNet-101 | SGD (freeze-embed) | **98.4** (+0.1) | 98.8 (-0.1) | 91.5 (-0.5) | 65.9 (-0.1) | 99.5 (+0.1) | 90.8 (-0.1) |
| BiT ResNet-101 | SGD (freeze-embed, no momentum) | 98.1 (-0.2) | **98.9** (-0.0) | 91.5 (-0.5) | 66.3 (+0.3) | 99.3 (-0.1) | **90.8** (-0.1) |

Table 2: **In-distribution (ID)** accuracies including SGD (freeze-embed) without momentum.

## 4.1 SoTA experiments and results

Our experiments get new state-of-the-art results for OOD accuracy on all 5 datasets. On 3/5 datasets (Living-17, WILDS-FMoW, and WILDS-Camelyon), our proposed SGD (freeze-embed) does the best, while in other 2, AdamW has a small edge. Here, state-of-the-art means that the numbers we get are better than, to our knowledge, any reported number and all numbers on the official leaderboard, and are better than standard full fine-tuning with SGD at the time of the preprint of the paper. We show the best results from our paper in Table 3 with a comparison to the previous state-of-the-art.

As a final point, we mention that if our hypothesis that AdamW would inherently avoid over-tuning of the embedding layer were true, unlike for SGD, freezing the embedding for AdamW would not be beneficial. In the Appendix, we expand the Tables 1-2 to include AdamW (freeze-embed) and indeed we see that freeze-embed does not provide complementary gains on top of AdamW.

|  | Living-17 | Waterbirds | DomainNet | FMoW | Camelyon |
|---|---|---|---|---|---|
| Best prior result | 87.6 | 89.3 | 87.2 | 47.6 | 93.3 |
| Best result from our paper | **90.5** | **89.8** | **93.8** | **49.9** | **96.5** |
| Optimizer for best result | SGD (freeze-embed) | AdamW | AdamW | SGD (freeze-embed) | SGD (freeze-embed) |
| Model for best result | CLIPViT-L/14 | ConvNeXt-B | CLIP ViT-L/14 | CLIP ViT-L/14 | CLIP ViT-L/14 |
| SGD (freeze-embed) result | **90.5** | 86.9 | 93.1 | **49.9** | **96.5** |

Table 3: Our OOD accuracy results compared with the best reported numbers in prior work on these datasets. We restrict to methods that *do not* use OOD data for hyperparameter selection or early stopping. To our knowledge, the previous state-of-the-art results are from Wortsman et al. (2022) for FMoW, Robey et al. (2021) for Camelyon, Kumar et al. (2022) for Living-17 and the version of DomainNet introduced by Tan et al. (2020), and Ghosal et al. (2022) for Waterbirds. The WILDS numbers and references are taken from the official WILDS leaderboard (as of 28 Sep 2022), and for Waterbirds we consider all methods that do not use group labels. For Living-17, we omit models pretrained with ImageNet-1K as Living-17 is a subset of ImageNet-1K.

# 5 DETAILED ANALYSIS OF CLIP.

CLIP models have strong transfer learning performance and robustness–among our 7 models, CLIP models had the best ID and OOD accuracies averaged across datasets. So we did a more detailed analysis of the CLIP ViT-B/16 with other optimizers[3] which are summarized in Table 5.

|  | CLIP ViT-B/16 | CLIP ViT-L/14 | CLIP ViT-H/14 |
|---|---|---|---|
| AdamW | 2.7 GB | 7.1 GB | Out-of-Memory |
| SGD | 2.3 GB | 6.1 GB | 10.1 GB |
| SGD (freeze-embed) | 2.3 GB | 6.1 GB | 10.1 GB |
| SGD (freeze-embed, no momentum) | **2.0 GB** | **4.8 GB** | **7.6 GB** |

Table 4: We profile the GPU memory consumption on three CLIP models of varying sizes, on a Titan-X GPU. SGD (freeze-embed) gives practical gains over AdamW, especially if we drop momentum. The largest CLIP ViT-H/14 does not fit in GPU memory when using AdamW, but fits in memory with other optimizers. Note that the freeze-embed methods perform competitively or better than AdamW on accuracy, as shown in Table 5.

**GPU memory profiling.** We profiled the GPU memory consumption of our 4 fine-tuning methods, on 3 models, namely CLIP-ViT B/16, CLIP ViT-L/14, and OpenCLIP ViT-H/14—the original CLIP model only scales up to ViT-L/14, so we used the OpenCLIP (Ilharco et al., 2021) for ViT-H/14. The profiling was done using Weights and Biases on a Titan-X GPU with micro-batch size of 1.

In Table 4, we see a ViT-B/16, AdamW uses $16\%$ and $36\%$ more memory than SGD (freeze-embed) and SGD (freeze-embed, no momentum), respectively. The gains are better for larger models: on a ViT-L/14, AdamW uses $18\%$ and $48\%$ more memory respectively. On a ViT-H/14, AdamW runs out of memory, while SGD (freeze-embed) and SGD (freeze-embed, no momentum) are able to run, showing that the gains are at least $20\%$ and $60\%$ respectively.

For large models, it is common to use additional tricks like gradient checkpointing to reduce the activation memory. That is, for a speed penalty (at most 2x), we only need to store activations in $\sqrt{L}$ of $L$ layers when doing backpropagation. Gradient checkpointing would further increase our gains over AdamW since they do not change the memory consumed by the weights but can substantially decrease the memory consumed by the model's activations.

**SGD (freeze-embed) and AdamW outperform LAMB and LARS.** We also ran two alternative adaptive gradient methods, LARS (You et al., 2017b) and LAMB (You et al., 2020)—also sweeping over 6 learning rates and early stopping. These are alternate methods with layerwise normalization that can avoid over-training of large gradient layers. Moreover, like SGD and SGD (freeze-embed), LARS also has a lower memory footprint than AdamW. In Table 5, we see that while LARS and LAMB get higher accuracies than SGD, they do worse than SGD (freeze-embed) and AdamW both ID and OOD. In this case, our modification with freeze-embed appears to be more effective.

---

[3]Generating Table 1 involved over 600 fine-tuning runs, so we weren't able to repeat this for every model.

| | Living-17 | | Waterbirds | | DomainNet | | FMoW | | Camelyon | | Avg. | |
| --- | --- | --- | --- | --- | --- | --- | --- | --- | --- | --- | --- | --- |
| | ID | OOD | ID | OOD | ID | OOD | ID | OOD | ID | OOD | ID | OOD |
| SGD | 97.8 (0.2) | 80.0 (1.3) | 97.2 (0.1) | 62.5 (5.0) | 88.8 (7.1) | 72.8 (18.0) | 67.0 (0.8) | 37.3 (1.1) | 99.4 (0.0) | 86.8 (1.1) | 90.0 | 67.9 |
| AdamW | 98.1 (0.1) | **82.8 (1.2)** | 97.7 (0.0) | **71.9 (2.4)** | 95.0 (0.1) | 89.2 (1.1) | **70.1 (0.2)** | **40.7 (0.3)** | **99.5 (0.0)** | **95.7 (0.4)** | **92.1** | **76.0** |
| SGD (freeze-embed) | **98.2 (0.3)** | 83.2 (0.8) | **97.8 (0.1)** | 73.7 (1.1) | **94.9 (0.3)** | 88.2 (0.7) | 70.0 (0.2) | 40.2 (0.7) | **99.5 (0.0)** | 94.3 (0.3) | **92.1** | 75.9 |
| SGD (freeze-em. no mom.) | **98.2** | 83.1 | 97.9 | 80.4 | **95.2** | 89.0 | **70.1** | 38.8 | **99.5** | 93.3 | **92.2** | 76.9 |
| LAMB | **98.2** | 79.5 | **97.8** | 64.0 | **95.1** | 90.4 | 67.9 | 38.8 | **99.5** | 93.4 | 91.7 | 73.2 |
| LARS | 97.7 | **83.9** | 97.1 | 48.6 | 93.2 | 83.8 | 67.0 | 38.6 | 99.3 | 93.3 | 90.9 | 69.6 |

Table 5: CLIP ViT-B/16 performance with new optimizers and confidence intervals. For addition to SGD, AdamW, and SGD (freeze-embed), we provide 90% confidence intervals based on 3 runs of each hyperparameter configuration. In addition, we show accuracies for *SGD (freeze-embed, no momentum)*; as well as comparison to two other optimizers, LARS (You et al., 2017b) and LAMB (You et al., 2020), which use layer-wise normalization in their updates.

**CIFAR-10 results.** As a proof of concept for standard (non-OOD) transfer learning, we fine-tune CLIP ViT models on CIFAR-10. Even at high accuracies, AdamW and SGD (freeze-embed) improve performance. SGD (freeze-embed) gets 20% and 30% lower error than SGD on CLIP ViT-B/16 and ViT-L/14, respectively. For example, on a CLIP ViT-L/14, SGD gets 99.0% accuracy but SGD (freeze-embed) gets 99.3% accuracy.

| | CLIP ViT-B/16 | CLIP ViT-L/14 |
| --- | --- | --- |
| SGD | 98.0 | 99.0 |
| AdamW | 98.3 | **99.3** |
| SGD (freeze-embed) | **98.4** | **99.3** |

Table 6: CIFAR-10 accuracy

**Composes with other fine-tuning methods.** In Appendix B we show that SGD (freeze-embed) can compose with other fine-tuning improvements such as LP-FT (Kumar et al., 2022) for further gains.

# 6 ADDITIONAL RELATED WORKS

Many works in transfer learning propose freezing parameters while fine-tuning to preserve pretrained information. For example, linear probing, which freezes the entire model except the head (Kumar et al., 2022; Wortsman et al., 2021) and zero-shot models (Radford et al., 2021) have been shown to improve OOD performance over standard fine-tuning. In NLP, methods such as prefix-tuning (Li & Liang, 2021) and prompt-tuning (Lester et al., 2021) have been shown to improve OOD accuracy. There are many other parameter efficient fine-tuning methods as well (Elsayed et al., 2019; Chen, 2023; Bahng et al., 2022; Tsao et al., 2023; Jia et al., 2022). Other works propose regularizing parameters towards initialization, freezing the first several layers, using different learning rates for different layers, or tuning different layers for different examples (Long et al., 2013; Ge & Yu, 2017; Howard & Ruder, 2018; Guo et al., 2019; Zhang et al., 2020; Zhu et al., 2020; Jiang et al., 2021; Aghajanyan et al., 2021). Typically, a large fraction of the model is frozen to preserve the pretrained information—a key difference in this work is that we find that freezing a very small fraction of the model (<1% of the parameters) can lead to substantial and consistent improvements in accuracy.

Other optimizers have been proposed to reduce AdamW's memory footprint, including LARS (You et al., 2017b) and AdaFactor (Shazeer & Stern, 2018). Our method is simpler and achieves better accuracies with same or better memory gains. A complementary line of work Dettmers et al. (2022) study quantization mechanisms for optimizer states. These tools, although developed for AdamW, can also be used with SGD (freeze-embed) to get additional gains in memory.

# 7 DISCUSSION.

We note that the methods we consider are not complex. We showed that a minor tweak of freezing the embedding layer overwhelmingly improves the performance across the board when fine-tuning with SGD. We clarify that we do not claim that SGD (freeze-embed) is a substantially better method than AdamW in terms of accuracy. Rather, it is remarkable that with its simplicity, we can already achieve *comparable* or even slightly better accuracy than AdamW across a wide range of models and benchmarks, while using much less memory. We hope future work can focus on extending these methods to other applications such as NLP, image segmentation, and recommender systems (Naumov et al., 2019), on models without embedding layers, and on more sophisticated learning rate schedules.

# 8 ACKNOWLEDGEMENTS

We thank Percy Liang, Tengyu Ma, Yuanzhi Li, and Zhiyuan Li for helpful comments.

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

## A  ADDITIONAL TRAINING DETAILS

**SGD and AdamW updates.**  We start with initialization $\theta^{(0)} = \theta^{\text{pretrain}}$. Given hyperparameters, minibatch size $batch\_size$ and number of epochs $num\_epochs$, our algorithms run for $T = \frac{num\_epochs \cdot |D_{\text{train}}|}{batch\_size}$ steps. At steps $t = 0, 1, \ldots, T$, we select a randomly shuffled minibatch $B_t$ from $D_{\text{train}}$ (reshuffled at end of each epoch) and compute the minibatch stochastic gradient $g_t = \frac{1}{|B_t|} \sum_{(x,y) \in B_t} \nabla_\theta l(f_\theta(x), y)$. The parameters $\theta^{(t)}$ are updated as follows.

For SGD, in addition to gradients $g_t$ and weights $\theta^{(t)}$, we maintain first order momentum estimate $m_t$ as optimizer state, initialized as $m_{-1} = 0$. The SGD($\eta_t, \mu, \lambda$) update with $\eta_t$ learning rate, $\mu$ momentum, and $\lambda$ weight decay, is given by

$$g_t = g_t + \lambda \theta^{(t)}$$
$$m_t = \mu m_{t-1} + g_t \quad \text{(first moment)}$$
$$\theta^{(t+1)} = \theta^{(t)} - \eta_t m_t$$

For AdamW, we maintain two additional optimizer states: the first moment estimate $m_t$ and second moment estimate $v_t$, initialized as $m_{-1} = v_{-1} = 0$. The AdamW($\eta_t, \beta_1, \beta_2, \lambda$) update with $\eta_t$ learning rate, $(\beta_1, \beta_2)$ betas, and $\lambda$ weight decay is given by

$$m_t = \beta_1 m_{t-1} + (1 - \beta_1) g_t \quad \text{(first moment)}$$
$$v_t = \beta_2 v_{t-1} + (1 - \beta_2) g_t^{\odot 2} \quad \text{(second moment)}$$
$$\widehat{m}_t = \frac{m_t}{(1 - \beta_1^t)}; \quad \widehat{v}_t = \frac{v_t}{(1 - \beta_2^t)}$$
$$\theta^{(t+1)} = (1 - \eta_t \lambda)\theta^{(t)} - \eta_t \frac{\widehat{m}_t}{\sqrt{\widehat{v}_t} + \epsilon}$$

**Gradual unfreezing.**  The success of freeze-embed for SGD is inline with our hypothesis that it is beneficial to change the lower layers less during fine-tuning. We could further refine this idea—instead of just freezing the first "embedding" layer, we could start by freezing the entire pretrained model (except for the head) and gradually unfreeze the rest of the model over training. Strategies like this have been explored by ULMFiT (Howard & Ruder, 2018) and others (Mukherjee & Awadallah, 2019; Romero et al., 2020).

For gradual-unfreezing, we found that naively using a single learning rate throughout this process or even a cosine learning rate schedule is not sufficient to achieve good performance. Instead, we fit an exponential learning rate decay law for our experiments (see details on the learning rate schedule below). The performance of gradual-unfreezing on CLIP ViT-B/16 is shown in Table 7 and 8. On some datasets, gradual-unfreezing performed particularly well. For example, it got an OOD accuracy of 93.2% on DomainNet and 96.5% on Camelyon—which are both competitive with SGD (freeze-embed) and AdamW on a much larger CLIP ViT-L/14 model. On average, it performed slightly worse than SGD (freeze-embed, no momentum). Overall , we find that the most consistent improvements come from simply freezing the embedding layer, which suggests that the embedding layer plays a particularly prominent role in modern vision architectures.

**Hyperparameter details**

• **Learning rate.**  We sweep over 6 learning rates for SGD ([3e-5, 1e-4, 3e-4, 1e-3, 3e-3, 1e-2]) and for AdamW ([3e-7, 1e-6, 3e-6, 1e-5, 3e-5, 1e-4]). As standard, we use smaller learning rates for AdamW because they work better. The best learning rate is chosen based on ID validation accuracy. For the LP-FT experiments which we run in Section B we divide all the learning rates by 10, since we have trained the linear probe, so the learning rate required to fine-tune the entire model is smaller.

• **Other optimizer parameters.**  We use all the default options for SGD and AdamW in pytorch except set the SGD momentum parameter to 0.9. The other default options are: for SGD, the weight decay is 0 and we use momentum without dampening, and for AdamW, the weight decay is 0.01, betas are (0.9,0.999) and epsilon is 1e-08.

- **Number of epochs.** We train for 20 epochs on Living-17, 20 epochs on Waterbirds, 50 epochs on DomainNet, 5 epochs on WILDS-FMoW, and 3 epochs on Camelyon. We use the same number of training epochs for all the optimizer methods we report results on.

- **Learning rate schedule.** With the exception of *Gradual-unfreezing* in Table 7 and 8, we use a cosine learning rate schedule and decay the starting learning rate to $0$ over the course of $T$ training steps. Learning rates schedule for *Gradual-unfreezing* is described below.

- **Learning rate schedule for gradual unfreezing.** For gradual unfreezing, we do not use a cosine learning rate scheduler. Instead, at epoch $t$ (0-indexed) out of $T$, we multiply the base learning rate by $\exp(3.73 * (1.0 - t/(T - 1)))$. This means we multiply the learning rate by $\exp(3.73) \approx 41.7$ in epoch $0$, and by $1$ in the last epoch. The intuition is that when we are tuning a smaller number of layers we need a higher learning rate than for full fine-tuning—for example, the optimal learning rates for head tuning is higher than the optimal learning rate for full fine-tuning. The exact constant (3.73) was selected by comparing the optimal learning rate for head tuning with full fine-tuning on Waterbirds (the optimal learning rate for head-tuning was approximately $\exp(3.73)$ larger for head-tuning). Without this decay schedule, for example using a vanilla cosine learning rate scheduler, gradual unfreezing worked similarly or worse than vanilla full fine-tuning.

- **Stopping criteria.** The results presented in the paper are from models early stopped based on ID validation accuracy. We sanity checked that the conclusions are similar if we use the last checkpoint.

**Data augmentation and input preprocessing.** Additionally, we use the following preprocessing and augmentations on our input images. We use very basic augmentations (e.g., only horizontal flips for WILDS, and standard augmentations from past work), including for our state-of-the-art results:

1. For WILDS-FMoW, the images are $224 \times 224$, so we do not resize, and only perform a random horizontal flip augmentation. We do not perform any augmentations at test time.

2. For WILDS-Camelyon, we resize the images to $224 \times 224$ with a bilinear interpolation (standard in pyorch), and only perform a random horizontal flip augmentation. We do not perform any augmentations at test time, just the resize.

3. For Living-17, we follow (Kumar et al., 2022) and perform a RandomResizedCrop to $224 \times 224$ sized images (using the default options in pyorch), and then a random horizontal flip augmentation while training. At test-time we resize the image to $256 \times 256$ and then take a centercrop of size $224 \times 224$.

4. For DomainNet, we follow (Kumar et al., 2022) and first resize the image to $256 \times 256$ with bicubic interpolation, then take a RandomCrop of size $224 \times 224$ (using the default options in pyorch), and then a random horizontal flip augmentation while training. At test-time we simply resize the image to $224 \times 224$ with bicubic interpolation.

5. For Waterbirds, we resize the image to $224 \times 224$ with bicubic interpolation and then take a centercrop of size $224 \times 224$. We apply the same transformation at test time.

**Embedding layer.** For SGD (freeze-embed), the exact layers we freeze are as follows:

1. CLIP ViTs: We freeze the patch-to-token embedding layer and layernorm.

2. Supervised and DINO ViTs: We freeze the patch-to-token embedding layer (there is no layernorm after this)

3. BiT-ResNets: We freeze the 'stem' and the first convolution block of the model. We tried freezing less and more of the model in our initial experiments, but it did not seem to help.

4. ConvNeXt-B: We freeze the 'stem' and the first stage of the model.

# B  ABLATIONS ON CLIP

In Tables 7-8, we show additional ablations for the CLIP ViT-B/16 model on all datasets.[4] We tried:

---

[4]Running each ablation for all models on all datasets is too computationally expensive.

1. SGD (freeze-embed, not layer-norm): For the CLIP model our freeze-embed variation freezes the bottom embedding layer along with the layernorm right after that. We ran an ablation where we only freeze the bottom linear embedding layer, but not the layer norm. This performs comparably with SGD (freeze-embed), which suggests that freezing the input layer is what's important, and the layer norm does not matter much.

2. SGD (5x lower LR on embed layer): Another idea from our analysis–where we found that the embedding layer seems to be why AdamW does better than SGD–is to use a smaller learning rate on embedding layer. Typically, this would involve additional hyperparameter tuning hence undesirable. However, as a heuristic, we ran SGD with 5x smaller learning rate for the embedding layer (since the gradients in the embedding layer are about 5x larger) compared to other layers. As expected, this improves over SGD, but does not do as well as SGD (freeze-embed).

3. SGD (no momentum): Since SGD (freeze-embed, no momentum) performed very well in our experiments, we also tried fine-tuning with full SGD (no freezing), but without momentum. We found that SGD (no momentum) and SGD perform comparable.

4. SGD (weight decay): Vanilla SGD is done without weight decay, but AdamW incorporates weight decay. We ran this ablation to confirm that the gains of AdamW are not because of weight decay. We used the torch SGD optimizer, and set the weight_decay argument to 0.01. Indeed, we found that SGD and SGD (weight decay) perform comparably, which suggests that weight_decay is not the reason for the improved performance of AdamW.

5. Linear probing: We freeze the pretrained model, and only train a linear probe on the features of the CLIP ViT-B/16 model. We train a logistic regression classifier using the sklearn library, sweeping over 50 regularization values in $\text{np.logspace}(-7, 2, 5)$

6. LP-FT: Kumar et al. (2022) show that first linear probing, and then full fine-tuning the entire model often works better, especially out-of-distribution. We run LP-FT as well, and for the full fine-tuning step we use SGD, AdamW, or SGD (freeze-embed), to test if our conclusions still hold with a better fine-tuning method. Note that the test accuracy on Waterbirds was a bit unstable early on so for this dataset we use the last epoch instead of early stopping on ID validation accuracy. Indeed, even with LP-FT, we find that AdamW slightly outperforms SGD out-of-distribution, and SGD (freeze-embed) outperforms both methods with an average accuracy of 77.5%. The accuracies with LP-FT are higher than regular fine-tuning, in line as Kumar et al. (2022).

| | Algorithms | Living-17 | Waterbirds | DomainNet | FMoW | Camelyon | Avg. |
|---|---|---|---|---|---|---|---|
| *Baselines* | SGD | 80.0 | 62.5 | 72.8 | 37.3 | 86.8 | 67.9 |
| | AdamW | 82.8 | 71.9 | 89.2 | 40.7 | 95.7 | 76.0 |
| *Our methods* | SGD (freeze-embed) | 83.2 | 73.7 | 88.2 | 40.2 | 94.3 | 75.9 |
| | SGD (freeze-embed, no momentum) | 83.1 | **80.4** | 89.0 | 38.8 | 93.3 | 76.9 |
| | Gradual-unfreezing | 81.9 | 69.1 | **93.2** | 40.5 | **96.5** | 76.2 |
| *Variations of SGD (freeze-embed)* | SGD (freeze-embed, not layer-norm) | 83.6 | 74.3 | 89.1 | 39.3 | 92.9 | 75.9 |
| | SGD (5x lower LR on embed layer) | 83.3 | 71.7 | 85.7 | 38.7 | 95.7 | 75.0 |
| | Other freezing: Linear probing | 86.2 | 60.4 | 89.1 | 29.0 | 92.6 | 71.5 |
| *Other layerwise normal-ization methods* | LAMB | 79.5 | 64.0 | 90.4 | 38.8 | 93.4 | 73.2 |
| | LARS | 83.9 | 48.6 | 83.8 | 38.6 | 93.3 | 69.6 |
| *Variations of SGD without any freezing* | SGD (no momentum) | 81.4 | 59.2 | 76.7 | 37.9 | 84.3 | 67.9 |
| | SGD (weight decay) | 83.9 | 65.1 | 67.5 | 37.1 | 85.6 | 67.8 |
| *Variations of LP-FT* | SGD | **86.7** | 67.3 | 89.2 | 37.9 | 94.1 | 75.0 |
| | AdamW | 84.5 | 68.2 | 90.1 | 39.7 | 95.8 | 75.7 |
| | SGD (freeze-embed) | 83.1 | 75.9 | 90.8 | **41.8** | 96.0 | **77.5** |

Table 7: **Out-of-distribution (OOD)** accuracy of more optimizers on CLIP ViT-B/16. We find that weight decay, momentum, and unfreezing the layer norm at the bottom of the model do not make much of a difference.

| | Algorithms | Living-17 | Waterbirds | DomainNet | FMoW | Camelyon | Avg. |
|---|---|---|---|---|---|---|---|
| *Baselines* | SGD | 97.8 | 97.2 | 88.8 | 67.0 | **99.4** | 90.0 |
| | AdamW | 98.1 | 97.7 | 95.0 | **70.1** | **99.5** | **92.1** |
| *Our methods* | SGD (freeze-embed) | **98.2** | 97.8 | 94.9 | **70.0** | **99.5** | **92.1** |
| | SGD (freeze-embed, no momentum) | **98.2** | 97.9 | 95.2 | **70.1** | **99.5** | **92.2** |
| | Gradual-unfreezing | **98.3** | **98.3** | **96.3** | 69.2 | 99.3 | **92.3** |
| *Variations of* | SGD (freeze-embed, not layer-norm) | 98.0 | 98.0 | 95.4 | **70.2** | **99.5** | **92.2** |
| *SGD (freeze-embed)* | SGD (5x lower LR on embed layer) | 98.0 | 97.5 | 94.8 | 68.7 | 99.5 | 91.7 |
| | Other freezing: Linear probing | 97.8 | 96.6 | 94.5 | 47.2 | 96.1 | 86.4 |
| *Other layerwise normal-* | LAMB | **98.2** | 97.8 | 95.1 | 67.9 | **99.5** | 91.7 |
| *ization methods* | LARS | 97.7 | 97.1 | 93.2 | 67.0 | 99.3 | 90.9 |
| *Variations of SGD* | SGD (no momentum) | 98.0 | 97.1 | 89.5 | 66.4 | 99.3 | 90.1 |
| *without any freezing* | SGD (weight decay) | 97.6 | 97.2 | 87.9 | 66.4 | 99.3 | 89.7 |
| *Variations of LP-FT* | SGD | **98.2** | 97.2 | 95.1 | 66.7 | 99.0 | 91.2 |
| | AdamW | **98.2** | **98.2** | 95.7 | 69.2 | **99.5** | **92.2** |
| | SGD (freeze-embed) | **98.4** | 97.8 | 95.7 | 69.1 | **99.4** | **92.1** |

Table 8: **In-distribution (ID)** accuracy of more optimizers on CLIP ViT-B/16. We find that weight decay, momentum, and unfreezing the layer norm at the bottom of the model do not make much of a difference.

## C    ABLATIONS ON FREEZING CONVNEXT

The convNeXt-Base model consists of a stem (6.5k parameters) then 4 stages (415 thousand, 1.7 million, 58 million, 27 million parameters respectively), followed by a layernorm (2k parameters). In the main paper, we freeze the stem and the first stage which consists of 0.5% of the parameters of the entire model. This is similar to the fraction of parameters in the patch embedding layer of the vision transformers, which we freeze.

An alternative choice is to freeze only the stem of the ConvNeXt layer, which is 0.007% of the parameters. We run an ablation where we try this choice of freezing, which we call SGD (freeze-stem). For our original approach of freezing the stem and the first block, we call it SGD (freeze-stem-block-1). Results for OOD are in Table 9, and for ID are in Table 10.

|                          | Living-17   | Waterbirds  | DomainNet  | FMoW        | Camelyon    | Avg.        |
|--------------------------|-------------|-------------|------------|-------------|-------------|-------------|
| SGD                      | **94.0**    | 80.2        | 89.8       | 39.7        | 83.0        | 77.3        |
| AdamW                    | 90.3 (-3.7) | **89.8** (+9.6) | 89.5 (-0.3) | 38.4 (-1.3) | **89.5** (+6.5) | **79.5** (+2.2) |
| SGD (freeze-stem-block-1)| 92.6 (-1.4) | 86.9 (+6.7) | **91.2** (+1.4) | 38.2 (-1.5) | 88.1 (+5.1) | **79.4** (+2.1) |
| SGD (freeze-stem)        | 91.4 (-2.6) | 84.7 (+4.5) | 91.0 (+1.2) | **40.3** (+0.6) | 86.2 (+3.2) | 78.7 (+1.4) |

Table 9: **Out-of-distribution (OOD)** accuracies for ConvNeXt where we try either freezing just the stem layer (0.5% of model parameters), or the stem and the first block (0.007% of model parameters).

|                          | Living-17   | Waterbirds  | DomainNet  | FMoW        | Camelyon    | Avg.        |
|--------------------------|-------------|-------------|------------|-------------|-------------|-------------|
| SGD                      | **98.7**    | 99.0        | 94.8       | 66.3        | 99.4        | 91.6        |
| AdamW                    | 98.6 (-0.1) | **99.5** (+0.5) | 94.5 (-0.3) | **68.8** (+2.5) | **99.7** (+0.3) | **92.2** (+0.6) |
| SGD (freeze-stem-block-1)| 98.6 (-0.1) | 99.4 (+0.4) | **95.1** (+0.3) | 67.4 (+1.1) | 99.5 (+0.1) | 92.0 (+0.4) |
| SGD (freeze-stem)        | **98.8** (+0.1) | 99.3 (+0.3) | **95.1** (+0.3) | 67.2 (+0.9) | 99.5 (+0.1) | 92.0 (+0.4) |

Table 10: **In-distribution (ID)** accuracies for ConvNeXt where we try either freezing just the stem layer (0.5% of model parameters), or the stem and the first block (0.007% of model parameters).

## D    RESULTS FOR ADAMW (FREEZE-EMBED)

In our paper, we hypothesized that AdamW and SGD (freeze-embed) improve on SGD for the same reason—they change the embedding layer less. Based on this hypothesis, we would expect that the gains of AdamW and freeze-embed are *not complementary*. Indeed, we find that the AdamW (freeze-embed) variation performs similarly to AdamW and SGD (freeze-embed).

Tables 11 & 12 expand on the OOD and ID accuracy results in Tables 11 & 12, respectively, to include AdamW (freeze-embed). Overall, AdamW (freeze-embed) and SGD (freeze-embed) perform comparably for all the recent vision models, both fairly close to AdamW—although there are differences in some models and datasets. Averaged across all the datasets and models, AdamW, AdamW (freeze-embed), and SGD (freeze-embed) get 76%, 76.5%, and 76.7% accuracy, respectively, compared to 72.0% for SGD. Averaged across all the *in-distribution* datasets and models, AdamW, AdamW (freeze-embed), and SGD (freeze-embed) get 91.5%, 91.5%, and 91.3% accuracy, respectively, compared to 90.3% for SGD. Overall the performances of the three models are similar enough to suggest that these methods work well for the same reason that they tune the embedding layers less, and that freeze-embed is not an independent axis of improvement.

| | | Living-17 | Waterbirds | DomainNet | FMoW | Camelyon | Avg. |
|---|---|---|---|---|---|---|---|
| CLIP ViT-B/16 | SGD | 80.0 | 62.5 | 72.8 | 37.3 | 86.8 | 67.9 |
| CLIP ViT-B/16 | AdamW | 82.8 (+2.8) | 71.9 (+9.4) | **89.2** (+16.4) | **40.7** (+3.4) | 95.7 (+8.9) | **76.0** (+8.1) |
| CLIP ViT-B/16 | SGD (freeze-embed) | **83.2** (+3.2) | **73.7** (+11.2) | 88.2 (+15.4) | 40.2 (+2.9) | 94.3 (+7.5) | 75.9 (+8.0) |
| CLIP ViT-B/16 | AdamW (freeze-embed) | 82.4 (+2.4) | 69.5 (+7.0) | 88.2 (+15.4) | **40.7** (+3.4) | **96.7** (+9.9) | 75.5 (+7.6) |
| CLIP ViT-L/14 | SGD | 84.2 | 65.0 | 60.8 | 41.0 | 83.2 | 66.8 |
| CLIP ViT-L/14 | AdamW | 88.0 (+3.8) | **85.2** (+20.2) | **93.8** (+33.0) | 48.3 (+7.3) | 95.9 (+12.7) | 82.2 (+15.4) |
| CLIP ViT-L/14 | SGD (freeze-embed) | **90.5** (+6.3) | 84.7 (+19.7) | 93.1 (+32.3) | **49.9** (+8.9) | **96.5** (+13.3) | **83.0** (+16.2) |
| CLIP ViT-L/14 | AdamW (freeze-embed) | 88.0 (+3.8) | 84.6 (+19.6) | **93.8** (+33.0) | 44.6 (+3.6) | 96.4 (+13.2) | 81.5 (+14.7) |
| Sup ViT-B/16 | SGD | **89.5** | 77.4 | **86.3** | 33.5 | 92.6 | 75.8 |
| Sup ViT-B/16 | AdamW | 88.3 (-1.2) | 81.6 (+4.2) | 84.4 (-1.9) | **35.9** (+2.4) | 87.9 (-4.7) | 75.6 (-0.2) |
| Sup ViT-B/16 | SGD (freeze-embed) | 88.0 (-1.5) | **82.4** (+5.0) | **86.3** (+0.0) | 34.4 (+0.9) | **93.7** (+1.1) | **77.0** (+1.2) |
| Sup ViT-B/16 | AdamW (freeze-embed) | 88.1 (-1.4) | **82.4** (+5.0) | 82.3 (-4.0) | 35.7 (+2.2) | 93.0 (+0.4) | 76.3 (+0.5) |
| DINO ViT-B/16 | SGD | **88.2** | 56.1 | 76.0 | 33.6 | 86.9 | 68.2 |
| DINO ViT-B/16 | AdamW | 87.4 (-0.8) | 61.2 (+5.1) | 77.4 (+1.4) | 35.8 (+2.2) | 91.9 (+5.0) | 70.7 (+2.5) |
| DINO ViT-B/16 | SGD (freeze-embed) | 86.7 (-1.5) | **67.9** (+11.8) | **78.4** (+2.4) | **35.9** (+2.3) | 90.6 (+3.7) | **71.9** (+3.7) |
| DINO ViT-B/16 | AdamW (freeze-embed) | 86.8 (-1.4) | 64.5 (+8.4) | 76.6 (+0.6) | 35.4 (+1.8) | **93.5** (+6.6) | 71.4 (+3.2) |
| ConvNext-Base | SGD | **94.0** | 80.2 | 89.8 | 39.7 | 83.0 | 77.3 |
| ConvNext-Base | AdamW | 90.3 (-3.7) | 89.8 (+9.6) | 89.5 (-0.3) | 38.4 (-1.3) | **89.5** (+6.5) | 79.5 (+2.2) |
| ConvNext-Base | SGD (freeze-embed) | 92.6 (-1.4) | 86.9 (+6.7) | **91.2** (+1.4) | 38.2 (-1.5) | 88.1 (+5.1) | 79.4 (+2.1) |
| ConvNext-Base | AdamW (freeze-embed) | 88.1 (-5.9) | **91.6** (+11.4) | 89.8 (+0.0) | **41.6** (+1.9) | 87.7 (+4.7) | **79.8** (+2.5) |
| BiT ResNet-50 | SGD | **84.3** | 76.5 | 80.0 | 34.1 | 90.4 | 73.1 |
| BiT ResNet-50 | AdamW | 83.1 (-1.2) | 74.8 (-1.7) | **84.0** (+4.0) | 33.8 (-0.3) | 92.4 (+2.0) | 73.6 (+0.5) |
| BiT ResNet-50 | SGD (freeze-embed) | 84.1 (-0.2) | 75.5 (-1.0) | 82.3 (+2.3) | 35.0 (+0.9) | **95.2** (+4.8) | 74.4 (+1.3) |
| BiT ResNet-50 | AdamW (freeze-embed) | 82.9 (-1.4) | **77.3** (+0.8) | 83.3 (+3.3) | **36.3** (+2.2) | 93.7 (+3.3) | **74.7** (+1.6) |
| BiT ResNet-101 | SGD | 82.8 | 76.9 | **86.2** | **38.0** | 89.3 | 74.6 |
| BiT ResNet-101 | AdamW | 82.9 (+0.1) | **79.4** (+2.5) | 83.5 (-2.7) | 37.0 (-1.0) | 89.7 (+0.4) | 74.5 (-0.1) |
| BiT ResNet-101 | SGD (freeze-embed) | 83.1 (+0.3) | 77.3 (+0.4) | 86.0 (-0.2) | 36.0 (-2.0) | 95.5 (+6.2) | 75.6 (+1.0) |
| BiT ResNet-101 | AdamW (freeze-embed) | **84.8** (+2.0) | 78.3 (+1.4) | 84.3 (-1.9) | **38.2** (+0.2) | 95.0 (+5.7) | **76.1** (+1.5) |

Table 11: **Out-of-distribution (OOD)** accuracies with AdamW (freeze-embed). This is an expansion of Table 1 to include AdamW (freeze-embed) OOD results for fine-tuning 7 popular models across 5 benchmark datasets. On OOD performance averaged across all models and datasets, AdamW (freeze-embed) gets slightly better accuracy than AdamW but slightly worse than SGD (freeze-embed).

| | | Living-17 | Waterbirds | DomainNet | FMoW | Camelyon | Avg. |
|---|---|---|---|---|---|---|---|
| CLIP ViT-B/16 | SGD | 97.8 | 97.2 | 88.8 | 67.0 | 99.4 | 90.0 |
| CLIP ViT-B/16 | AdamW | **98.1** (+0.3) | **97.7** (+0.5) | 95.0 (+6.2) | 70.1 (+3.1) | **99.5** (+0.1) | **92.1** (+2.1) |
| CLIP ViT-B/16 | SGD (freeze-embed) | 98.2 (+0.4) | 97.8 (+0.6) | 94.9 (+6.1) | 70.0 (+3.0) | **99.5** (+0.1) | 92.1 (+2.1) |
| CLIP ViT-B/16 | AdamW (freeze-embed) | **98.3** (+0.5) | **97.8** (+0.6) | **95.3** (+6.5) | **70.5** (+3.5) | **99.6** (+0.2) | **92.3** (+2.3) |
| CLIP ViT-L/14 | SGD | 98.4 | 97.3 | 84.3 | 69.0 | 99.4 | 89.7 |
| CLIP ViT-L/14 | AdamW | **98.9** (+0.5) | **98.8** (+1.5) | 96.9 (+12.6) | 74.5 (+5.5) | **99.6** (+0.2) | **93.7** (+4.0) |
| CLIP ViT-L/14 | SGD (freeze-embed) | 98.7 (+0.3) | **98.9** (+1.6) | **97.1** (+12.8) | 74.5 (+5.5) | **99.6** (+0.2) | **93.7** (+4.0) |
| CLIP ViT-L/14 | AdamW (freeze-embed) | 98.7 (+0.3) | 98.8 (+1.5) | 96.7 (+12.4) | **75.1** (+6.1) | **99.6** (+0.2) | **93.8** (+4.1) |
| Sup ViT-B/16 | SGD | 98.6 | **99.1** | 91.7 | 64.1 | 99.4 | 90.6 |
| Sup ViT-B/16 | AdamW | **98.7** (+0.1) | **99.0** (-0.1) | **91.7** (-0.0) | 66.4 (+2.3) | **99.5** (+0.1) | **91.1** (+0.5) |
| Sup ViT-B/16 | SGD (freeze-embed) | **98.7** (+0.1) | **99.2** (+0.1) | 91.5 (-0.2) | 65.0 (+0.9) | **99.6** (+0.2) | 90.8 (+0.2) |
| Sup ViT-B/16 | AdamW (freeze-embed) | 98.6 (+0.0) | **99.0** (-0.1) | 90.9 (-0.8) | **66.8** (+2.7) | **99.5** (+0.1) | **91.0** (+0.4) |
| DINO ViT-B/16 | SGD | **98.4** | 97.0 | 88.2 | 62.4 | 99.4 | 89.1 |
| DINO ViT-B/16 | AdamW | **98.5** (+0.1) | **97.9** (+0.9) | 89.4 (+1.2) | 66.0 (+3.6) | **99.6** (+0.2) | **90.3** (+1.2) |
| DINO ViT-B/16 | SGD (freeze-embed) | **98.4** (+0.0) | 97.5 (+0.5) | 89.0 (+0.8) | 63.5 (+1.1) | **99.5** (+0.1) | 89.6 (+0.5) |
| DINO ViT-B/16 | AdamW (freeze-embed) | **98.4** (+0.0) | 97.8 (+0.8) | **89.7** (+1.5) | 65.7 (+3.3) | **99.6** (+0.2) | **90.3** (+1.2) |
| ConvNext-Base | SGD | **98.7** | 99.0 | 94.8 | 66.3 | 99.4 | 91.6 |
| ConvNext-Base | AdamW | 98.6 (-0.1) | **99.5** (+0.5) | 94.5 (-0.3) | 68.8 (+2.5) | **99.7** (+0.3) | 92.2 (+0.6) |
| ConvNext-Base | SGD (freeze-embed) | 98.6 (-0.1) | 99.4 (+0.4) | **95.1** (+0.3) | 67.4 (+1.1) | **99.5** (+0.1) | 92.0 (+0.4) |
| ConvNext-Base | AdamW (freeze-embed) | **98.7** (+0.0) | **99.5** (+0.5) | 94.7 (-0.1) | **69.2** (+2.9) | **99.6** (+0.2) | **92.4** (+0.8) |
| BiT ResNet-50 | SGD | 97.4 | **98.4** | 89.3 | 64.6 | 99.5 | **89.8** |
| BiT ResNet-50 | AdamW | 97.2 (-0.2) | **98.5** (+0.1) | 89.2 (-0.1) | **65.1** (+0.5) | **99.5** (+0.0) | **89.9** (+0.1) |
| BiT ResNet-50 | SGD (freeze-embed) | 97.6 (+0.2) | **98.5** (+0.1) | 89.2 (-0.1) | 64.8 (+0.2) | **99.5** (+0.0) | **89.9** (+0.1) |
| BiT ResNet-50 | AdamW (freeze-embed) | 97.4 (+0.0) | 98.4 (-0.0) | **89.1** (-0.2) | 64.9 (+0.3) | **99.6** (+0.1) | **89.9** (+0.1) |
| BiT ResNet-101 | SGD | 98.3 | 98.9 | 92.0 | 66.0 | 99.4 | 90.9 |
| BiT ResNet-101 | AdamW | **98.4** (+0.1) | 98.6 (-0.3) | 91.1 (-0.9) | **67.0** (+1.0) | **99.6** (+0.2) | **90.9** (-0.0) |
| BiT ResNet-101 | SGD (freeze-embed) | **98.4** (+0.1) | **98.8** (-0.1) | 91.5 (-0.5) | 65.9 (-0.1) | **99.5** (+0.1) | 90.8 (-0.1) |
| BiT ResNet-101 | AdamW (freeze-embed) | **98.2** (-0.1) | 98.7 (-0.2) | 91.1 (-0.9) | 66.6 (+0.6) | **99.6** (+0.2) | **90.9** (-0.0) |

Table 12: **In-distribution (ID)** accuracies with AdamW (freeze-embed). This is an expansion of Table 2 to include AdamW (freeze-embed) ID results for our 7 models and 5 datasets. AdamW, AdamW (freeze-embed), and SGD (freeze-embed) all perform comparably on ID accuracies.

## E  WHAT COULD CAUSE LARGE "EMBEDDING" LAYER GRADIENTS?

Generally speaking, we find that AdamW fine-tuning leads to better performing and more robust models than SGD fine-tuning, specially in modern pretrained models. Our algorithms to close this gap were inspired by the observation that on modern vision pretrained models, the embedding layers have substantially larger gradients at pretrained initialization compared to other layers. This could lead to over-training of the embedding layer when using SGD, which we hypothesized would be bad for robust fine-tuning. The success of SGD (freeze-embed) in our experiments adds further evidence to our hypothesis. But, why are the gradients of embedding layers at pretrained initialization high in the first place? More generally, why does AdamW do better than SGD during fine-tuning? We discuss two plausible hypotheses and then test these out in a controlled experiment.

**Algorithmic aspects: pretraining algorithm?**   Among our 7 models, the more recent models like vision transformers and ConvNeXt, which were pretrained with AdamW are also the ones with largest gaps in performance between AdamW and SGD fine-tuning. BiT ResNets that were pretrained with SGD had much smaller differences between AdamW and SGD. This strongly suggests that the discrepancy between the algorithms in pretraining and fine-tuning might be cause for the performance gaps. For example, it is possible that pretraining with AdamW implicitly biases towards configurations that most benefit from AdamW updates. This would also explain why other adaptive algorithms like LARS and LAMB are not competitive even though they perform some form of layer-wise normalization. On the other hand it does not explain why such effects would distinctively impact the "embedding" layer and not the other layers. In a related manner, it is also unclear why SGD (freeze-embed) would be able to overcome such implicit biases from AdamW pretraining.

**Architectural aspects: "patchifying" first layer?**   There are substantial architectural differences between the newer transformer and ConvNeXt models and the older ResNets which could also contribute to the newer models working better with AdamW. Most notably, vision transformer is fundamentally different architecture from convolutional networks. At the same time, the biggest differences between the architectures—like self-attention, softmax non-linearity, or fully connected layers—are *not* the likely contributors to the differences between AdamW and SGD in fine-tuning. This is because, we also see gaps between these methods with the ConvNeXt models which lack the major transformer components. Rather, it is the possible that some of the designs that were adapted from transformers to ConvNeXt contributes to the differences between AdamW and SGD fine-tuning. Among these, many primarily affect the higher layers such as heavy third stage, depthwise convolution, inverted bottleneck, and larger kernels, and are unlikely to cause the lower layer gradients to be high. The key design change we believe is important here is the use of a "patchify stem" that could cause distinctive changes in the gradients for lower blocks.

The "stem" layer primarily controls how the input is processed for the rest of the network. ViTs and ConvNext down-samples the input images from non-overlapping patch tokens, compared to ResNets that use denser-overlapped convolution followed by max-pool. The coarser non-overlap might lead the embedding layer to attune more closely to the patch distribution in pretraining. This might not be an issue by itself as pixel/patch level information is noisy anyways and the higher layers can extract more robust features from across the patches. A possible dynamics in fine-tuning is as follows: On new datasets where the patch distributions are very different from pretraining, the embedding layers might have large gradients. In standard SGD, this might cause the embedding layer to moving too quickly to fit the new patches before the higher layers can adapt to the changes. Instead, freezing the embedding layer would pass along the raw patches with minimal processing and lets the model adapt to the new distribution based on higher level features, which are likely more robust.

**Summary of hypotheses.**   In conclusion, we hypothesize that the differences between AdamW and SGD fine-tuning arise from a combination of the above reasons—pretraining with AdamW and large "patchify" embedding layer. Additionally, the use of GeLU activation and layer normalization might also change the optimizer dynamics although these are minor changes from ReLU activation and group normalization used in BiT ResNets. It is of interest to systematically explore these reasons further in future work.

**Controlled experiments to test hypotheses.**   We test these hypotheses out in controlled experiments and find that the mismatch between the pretraining and fine-tuning optimization method seems

to be the key factor (for both why SGD can underperform during fine-tuning, and why the first-layer gradients of modern pretrained models is higher).

To test this out, we pretrained a BiT-ResNet-50 model using either (a) SGD or (b) AdamW. For each of these pretraining optimizer choices, we also tried using either (i) the original BiT-ResNet-50 architecture or (ii) modifying it by "patchifying" the first layer. This gives us 4 pretrained models. For each of these pretrained models, we tried fine-tuning it with (1) SGD, (2) AdamW, and (3) SGD-Freeze-Embed.Note that for ResNet models we consider the stem and the first stage as part of the 'embedding' layer.

For the AdamW pretrained model, fine-tuning with SGD does worse than AdamW on the OOD test sets (with or without the patchify architecture change), but for the SGD pretrained models fine-tuning with SGD does slightly better than AdamW. For example, for the standard BiT ResNet-50 architecture pretrained with AdamW, fine-tuning with AdamW gets an average 1.3% higher accuracy OOD. For the BiT ResNet-50 (patchify) architecture pretrained with AdamW, fine-tuning with AdamW gets an average 0.9% higher accuracy OOD. On the other hand, for both ResNet models pretrained with SGD, fine-tuning with AdamW does slightly *worse* than SGD. This suggests that the optimizer mismatch (between pretraining and fine-tuning) is a key reason for why fine-tuning with SGD can do worse than fine-tuning with AdamW. Nonetheless, fine-tuning with SGD (freeze-embed) consistently performs well, getting the highest average OOD accuracy in 3/4 cases (its OOD accuracy is in between SGD and AdamW on a ResNet patchify model pretrained with SGD).

The full results for OOD are in Table 13 and for ID are in Table 14. As in the main paper, the ID accuracies of all methods are fairly close.

| Pretraining | Fine-tuning | Living-17 | Waterbirds | DomainNet | FMoW | Camelyon | Avg. |
|---|---|---|---|---|---|---|---|
| SGD | SGD | 83.9 | 63.4 | 77.9 | **34.2** | 88.3 | 69.6 |
| SGD | AdamW | 77.5 (-6.4) | **66.4** (+3.0) | 75.5 (-2.4) | 33.4 (-0.8) | 94.6 (+6.3) | 69.5 (-0.1) |
| SGD | SGD (freeze-embed) | **84.2** (+0.3) | 65.3 (+1.9) | **79.3** (+1.4) | 32.2 (-2.0) | **95.6** (+7.3) | **71.3** (+1.7) |
| AdamW | SGD | **81.4** | 56.9 | 74.4 | **30.7** | 86.6 | 66.0 |
| AdamW | AdamW | 80.1 (-1.3) | **65.6** (+8.7) | 73.6 (-0.8) | **30.6** (-0.1) | 86.6 (+0.0) | 67.3 (+1.3) |
| AdamW | SGD (freeze-embed) | 81.1 (-0.3) | 60.9 (+4.0) | **77.5** (+3.1) | 30.3 (-0.4) | **89.6** (+3.0) | **67.9** (+1.9) |
| SGD (Patchify) | SGD | 82.5 | 67.0 | 78.6 | 35.2 | 91.0 | **70.8** |
| SGD (Patchify) | AdamW | 79.8 (-2.7) | 64.3 (-2.7) | 75.2 (-3.4) | 34.6 (-0.6) | 90.9 (-0.1) | 69.0 (-1.8) |
| SGD (Patchify) | SGD (freeze-embed) | 82.5 (+0.0) | 62.9 (-4.1) | 78.5 (-0.1) | 34.9 (-0.3) | **91.2** (+0.2) | 70.0 (-0.8) |
| AdamW (Patchify) | SGD | 81.3 | 61.7 | 74.5 | 31.7 | 93.3 | 68.5 |
| AdamW (Patchify) | AdamW | 79.3 (-2.0) | **69.5** (+7.8) | 73.1 (-1.4) | **33.2** (+1.5) | 91.9 (-1.4) | 69.4 (+0.9) |
| AdamW (Patchify) | SGD (freeze-embed) | **82.0** (+0.7) | 64.0 (+2.3) | **77.0** (+2.5) | 31.1 (-0.6) | **95.0** (+1.7) | **69.8** (+1.3) |

Table 13: **Out-of-distribution (OOD)** accuracies for the different BiT-ResNet pretrained models.

| Pretraining | Fine-tuning | Living-17 | Waterbirds | DomainNet | FMoW | Camelyon | Avg. |
|---|---|---|---|---|---|---|---|
| SGD | SGD | **97.6** | **97.6** | **88.3** | **63.1** | **99.5** | **89.2** |
| SGD | AdamW | **97.8** (+0.2) | **97.6** (-0.0) | 87.2 (-1.1) | 62.4 (-0.7) | **99.5** (+0.0) | 88.9 (-0.3) |
| SGD | SGD (freeze-embed) | **97.6** (+0.0) | **97.6** (-0.0) | 87.3 (-1.0) | 62.7 (-0.4) | **99.5** (+0.0) | 89.0 (-0.2) |
| AdamW | SGD | 97.2 | 97.2 | **87.5** | **58.9** | 99.3 | **88.0** |
| AdamW | AdamW | **97.4** (+0.2) | **97.3** (+0.1) | 86.5 (-1.0) | 58.1 (-0.8) | **99.5** (+0.2) | 87.8 (-0.2) |
| AdamW | SGD (freeze-embed) | **97.4** (+0.2) | 97.1 (-0.1) | 86.9 (-0.6) | 58.2 (-0.7) | 99.3 (+0.0) | 87.8 (-0.2) |
| SGD (Patchify) | SGD | 97.9 | 97.5 | **88.3** | 62.8 | 99.4 | **89.2** |
| SGD (Patchify) | AdamW | 97.5 (-0.4) | 97.3 (-0.2) | 87.0 (-1.3) | 62.2 (-0.6) | **99.5** (+0.1) | 88.7 (-0.5) |
| SGD (Patchify) | SGD (freeze-embed) | 97.6 (-0.3) | 97.5 (-0.0) | 86.6 (-0.7) | **63.1** (+0.3) | 99.4 (-0.0) | 89.0 (-0.2) |
| AdamW (Patchify) | SGD | 97.5 | 97.3 | **87.7** | 59.6 | 99.3 | **88.3** |
| AdamW (Patchify) | AdamW | **97.6** (+0.1) | 97.2 (-0.1) | 86.2 (-1.5) | 58.4 (-1.2) | **99.5** (+0.2) | 87.8 (-0.5) |
| AdamW (Patchify) | SGD (freeze-embed) | **97.6** (+0.1) | 97.2 (-0.1) | 87.0 (-0.7) | 59.4 (-0.2) | 99.2 (-0.1) | **88.1** (-0.2) |

Table 14: **In-distribution (ID)** accuracies for the different BiT-ResNet pretrained models.

As in Section 3.2 we also plot the average gradient norm at each layer, across minibatches of the dataset, as described in Equation 3.1. We show the plot for DomainNet, Waterbirds, and Living-17 in Figure 3. We also show plots for alternative ways of normalizing the gradients in Figure 4 and Figure 5. For *SGD pretrained* ResNet, the *embedding layer has lower gradient* than the other layers. For *AdamW pretrained* ResNet, the *embedding layer has higher gradient* than the other layers. In

other words, pretraining with SGD versus with AdamW leads to a substantial difference in the embedding layer. Using a 'patchify' stem does not substantially change these results.

These results provide further evidence that (1) SGD does worse than AdamW when fine-tuning Vision Transformer and ConvNeXt models because these models are pretrained using AdamW, (2) the embedding layer plays a key role here, since pretraining with different optimizers leads to very different behavior at the embedding layer, (3) SGD (freeze-embed) can potentially improve fine-tuning, by freezing the embedding layer. Without this, SGD either over-optimizers the embedding layer (if the learning rate is large), or under-optimizers the other layers (if the learning rate is small).

## F    ADDITIONAL PLOTS ON GRADIENT NORMS AT THE PRETRAINED INITIALIZATION

In Figure 6 we show plots for the gradients norms at different layers for Living-17 and Waterbirds. These are analogous to Figure 2 for DomainNet in the main paper. We again see that, among the pretrained layers, the embedding layer has much higher gradients in the modern architectures.

We also consider two ways of normalizing the gradient magnitude. In the first method, we divide the norm of the gradient by the norm of the parameters in order to capture the relative "movement" in the first SGD step as opposed to the absolute "movement" which is captured by gradient norm itself. In the second method, we divide the norm of the gradient by the square root of the number of parameters. This is to check that a layer does not simply have a larger gradient because it has more parameters. The reason we use square root is as follows: suppose each parameter has gradient $\approx c$, then the layerwise gradient norm scales with the square root of the number of parameters. Also, the first step of AdamW update is essentially a signed gradient descent step, wherein if we ignore weight decay, the per-layer "movement" is the square root of the number of parameters. So this normalization can be thought of as relative size of SGD update compared to AdamW in each layer at initialization. For visualization purposes, we exclude the 'head' layer gradient in these plots as they often much larger than the others so the plots become hard to see if we include the 'head' layer. Note that we expect the head layer to have higher gradients because it is randomly initialized (Kumar et al., 2022). For ViT models, we omit gradients of the cls token, position embedding, and layer norm after the embedding layer.

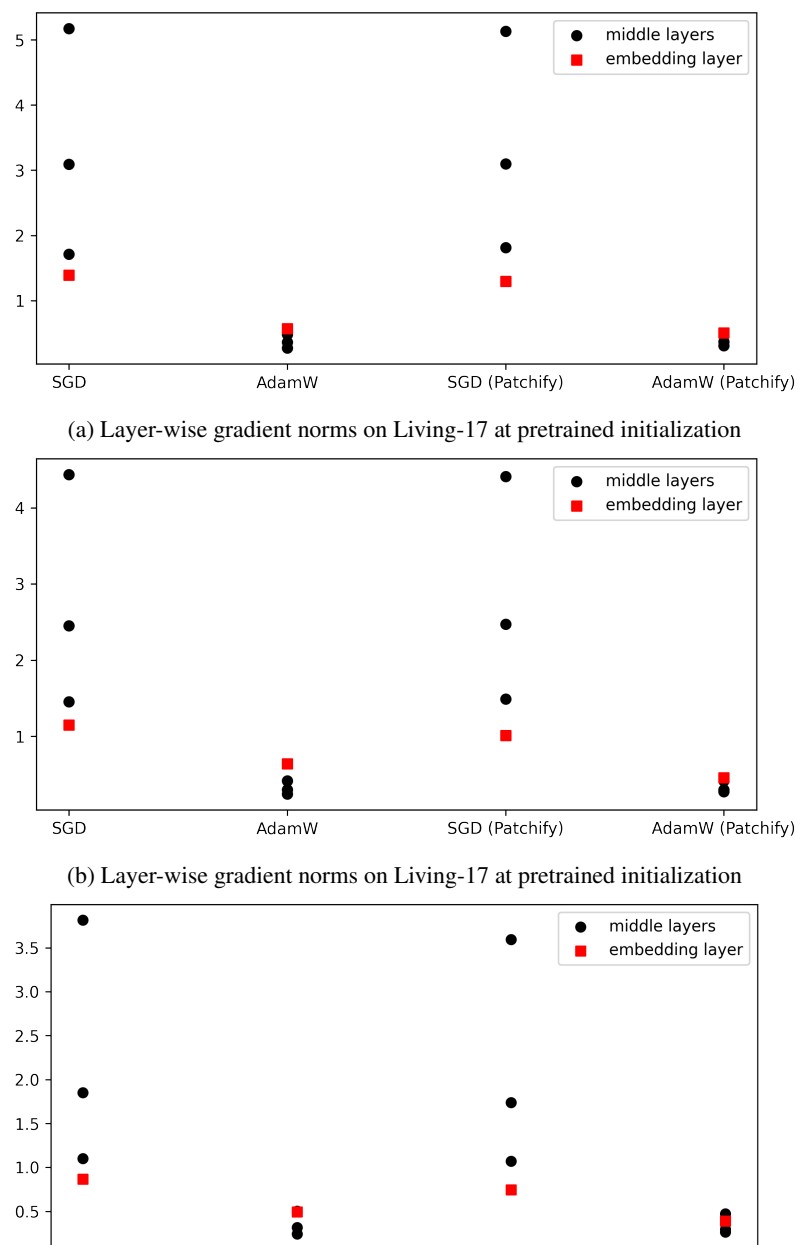

(a) Layer-wise gradient norms on Living-17 at pretrained initialization

(b) Layer-wise gradient norms on Living-17 at pretrained initialization

(c) Layer-wise gradient norms on Waterbirds at pretrained initialization

Figure 3: We visualize the layer-wise gradient norms of the four Bit-ResNet models on (a) DomainNet, (b) Living-17 and (c) Waterbirds, at the pretrained initialization. For better visualization, we omit the head from the plot, which predictably has much larger gradients than the others (since it is randomly initialized). The format is the same as Figure 2: gradient norms of "embedding" and "middle" layers are shown as **red-squares** and **black-circles**, respectively. We see that the "embedding" layer has higher gradient (than the other layers) for models pretrained with AdamW, but lower gradient (than the other layers) for models pretrained with SGD, which supports the hypotheses that the 'embedding' layer plays a key role, and that pretraining with AdamW vs. SGD leads to very different models and is responsible for this behavior.

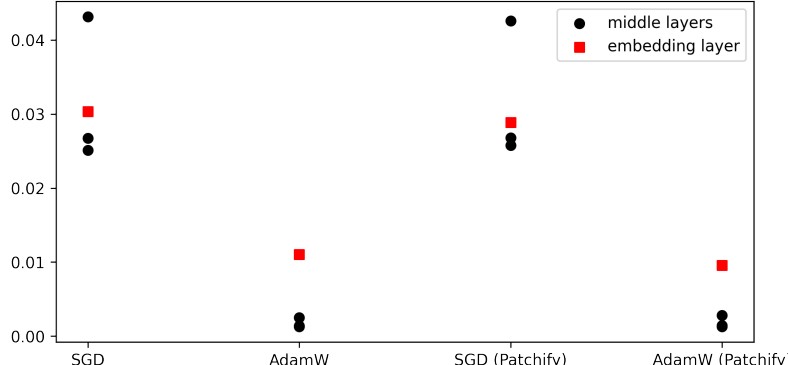

(a) Layer-wise gradient norms divided by parameter norm, on DomainNet at pretrained initialization

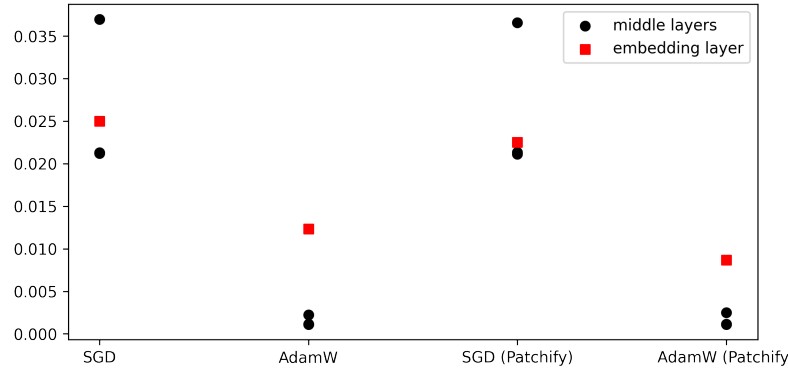

(b) Layer-wise gradient norms divided by parameter norm, on Living-17 at pretrained initialization

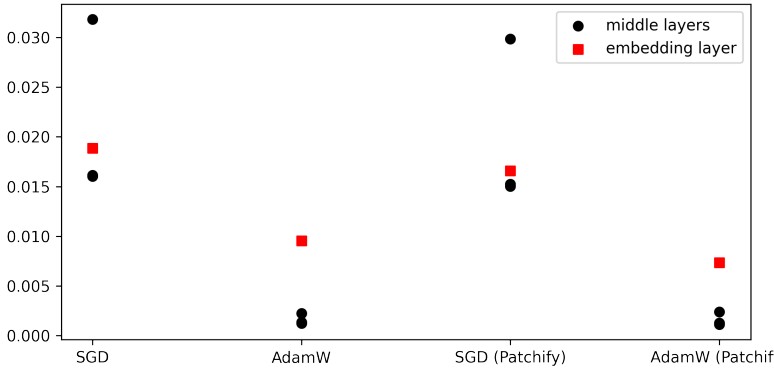

(c) Layer-wise gradient norms divided by parameter norm, on Waterbirds at pretrained initialization

Figure 4: We visualize the layer-wise gradient norm, **divided by the norm of the parameters** on (a) DomainNet, (b) Living-17, and (c) Waterbirds, at the pretrained initialization. For better visualization, we omit the head from the plot, which predictably has much larger gradients than the others (since it is randomly initialized). The format is the same as Figure 2: gradient norms of "embedding" and "middle" layers are shown as **red-squares** and **black-circles**, respectively. Under this normalization scheme, the "embedding" layer has much higher gradients when pretrained with AdamW, but the "embedding" layer gradient is somewhere in the middle for models pretrained with SGD.

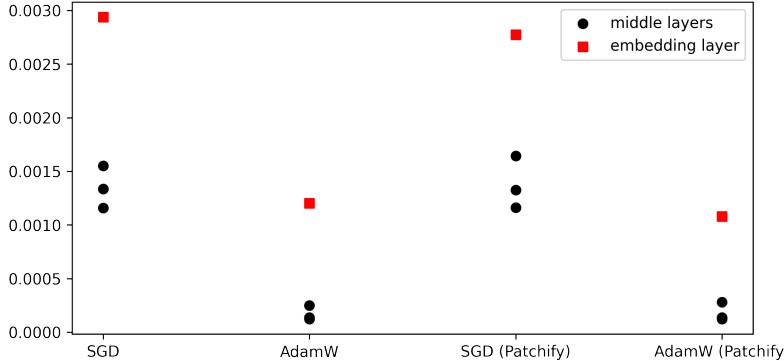

(a) Layer-wise gradient norms divided by $\sqrt{\text{#parameters}}$, on DomainNet at pretrained initialization

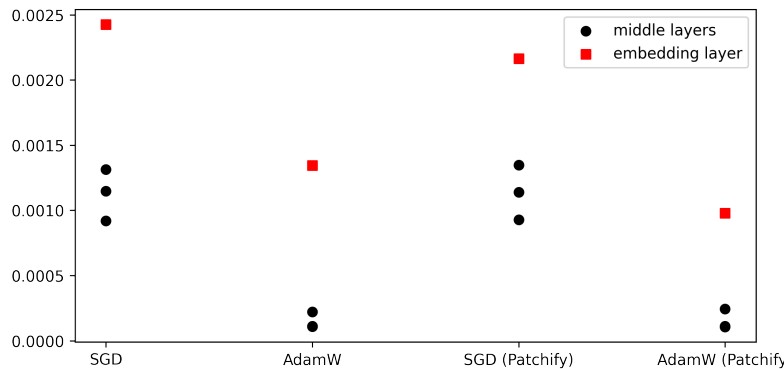

(b) Layer-wise gradient norms divided by $\sqrt{\text{#parameters}}$, on Living-17 at pretrained initialization

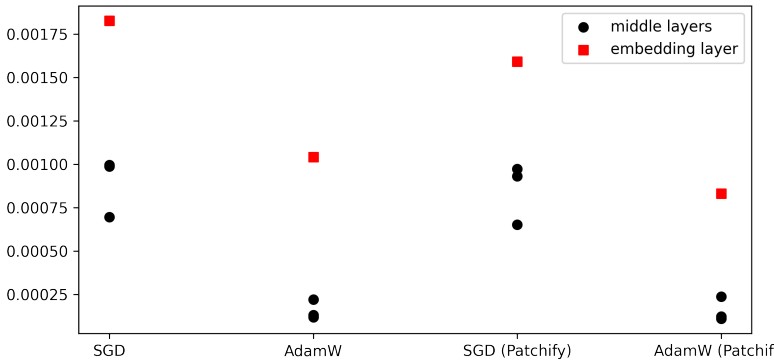

(c) Layer-wise gradient norms divided by $\sqrt{\text{#parameters}}$, on Waterbirds at pretrained initialization

Figure 5: We visualize the layer-wise gradient norm, **divided by the square root of the number of parameters** on (a) DomainNet, (b) Living-17, and (c) Waterbirds, at the pretrained initialization. For better visualization, we omit the head from the plot which has predictably much larger than the others (since it is randomly initialized). The format is the same as Figure 2: gradient norms of "embedding" and "middle" layers are shown as **red-squares** and **black-circles**, respectively. Under this normalization scheme, the "embedding" layer has higher gradients in all cases. However, the embedding layer gradient is about 2-3 times larger (than other layers) for models pretrained with SGD, but over 10 times larger (than other layers) for models pretrained with AdamW.

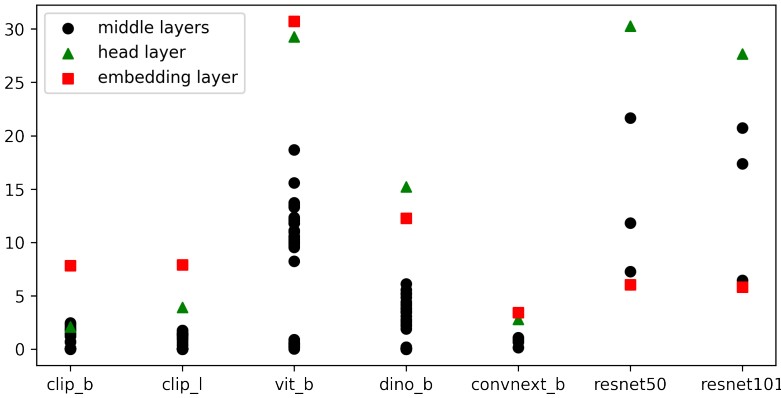

(a) Layer-wise gradient norms on Living-17 at pretrained initialization

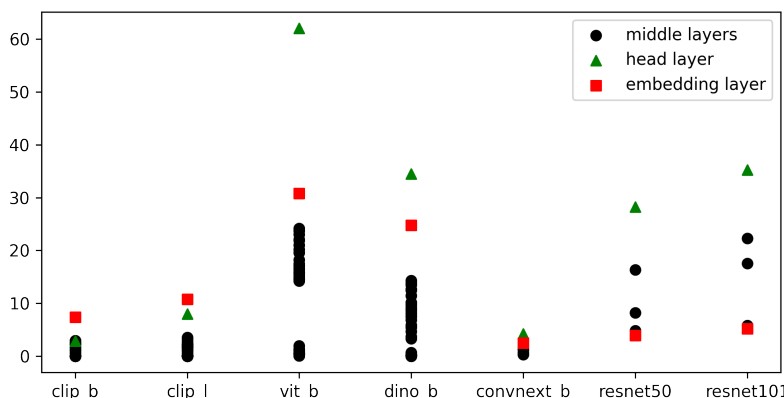

(b) Layer-wise gradient norms on Waterbirds at pretrained initialization

Figure 6: We visualize the layer-wise gradient norms our models on (a) Living-17 and (b) Waterbirds, at the pretrained initialization. The format is the same as Figure 2: gradient norms of "embedding", "head", and "middle" layers are shown as **red-squares**, **green-triangles** and **black-circles**, respectively.

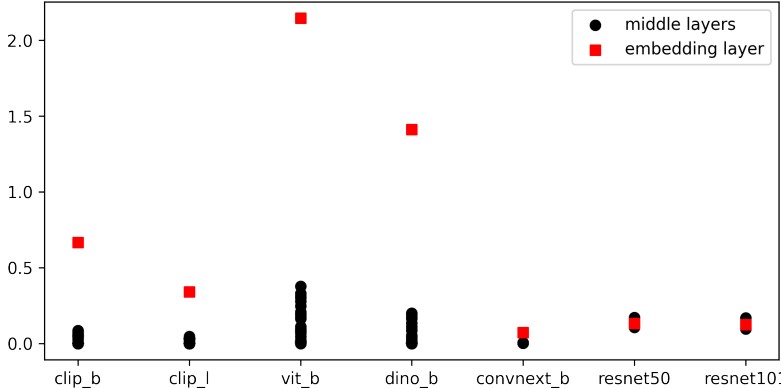

(a) Layer-wise gradient norms divided by parameter norm, on DomainNet at pretrained initialization

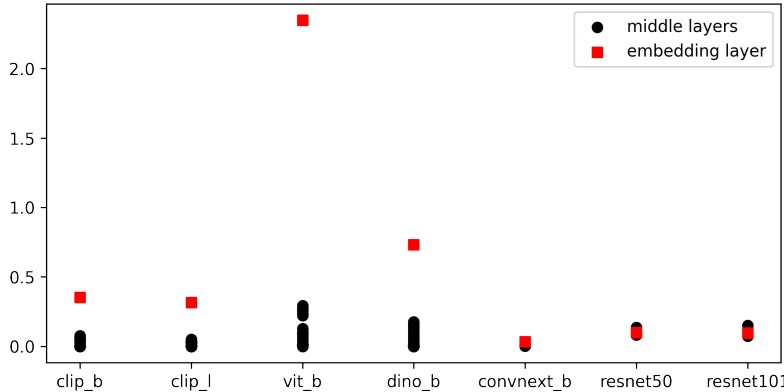

(b) Layer-wise gradient norms divided by parameter norm, on Living-17 at pretrained initialization

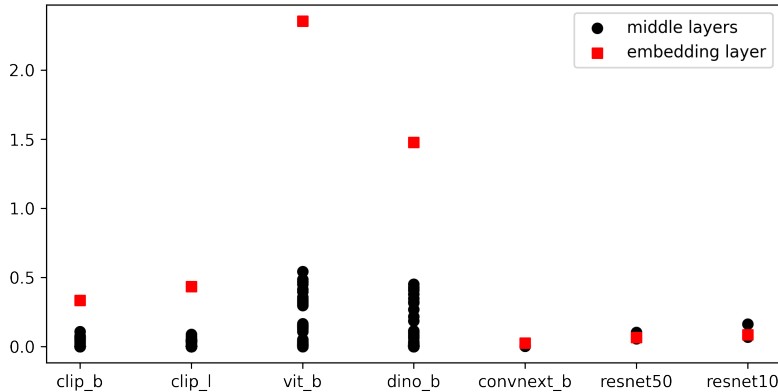

(c) Layer-wise gradient norms divided by parameter norm, on Waterbirds at pretrained initialization

Figure 7: We visualize the layer-wise gradient norm, **divided by the norm of the parameters** on (a) DomainNet, (b) Living-17, and (c) Waterbirds, at the pretrained initialization. For better visualization, we omit the head from the plot, which predictably has much larger gradients than the others (since it is randomly initialized). The format is the same as Figure 2: gradient norms of "embedding" and "middle" layers are shown as **red-squares** and **black-circles**, respectively. Under this normalization scheme, the embedding layer has higher gradients than the other layers in all models. However, the gradient is only slightly larger for ResNet models, and substantially larger for the Vision Transformer models—which also provides support for why freezing the embedding layer in Vision Transformers might make a larger difference.

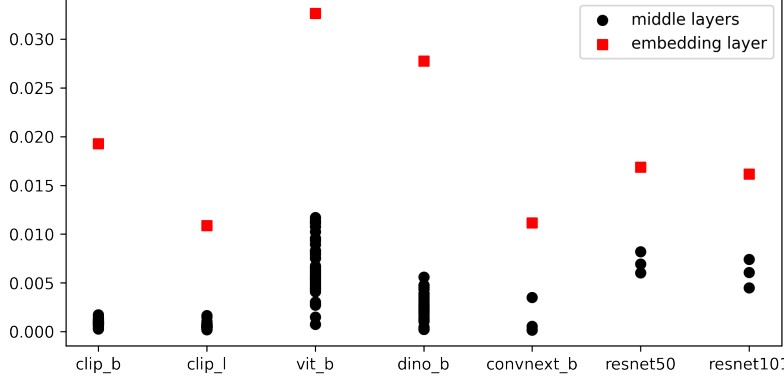

(a) Layer-wise gradient norms divided by $\sqrt{\#\text{parameters}}$, on DomainNet at pretrained initialization

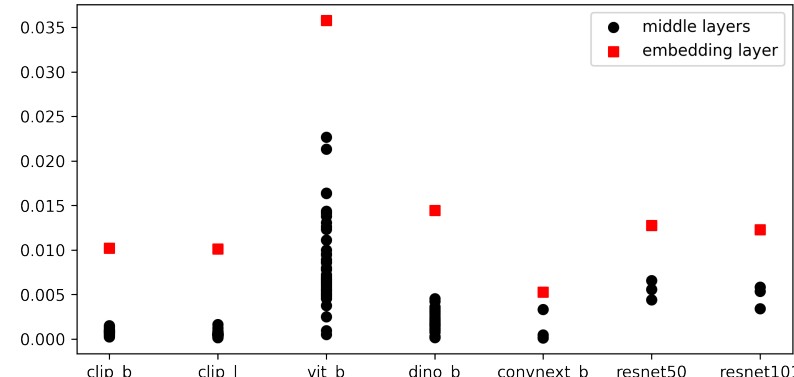

(b) Layer-wise gradient norms divided by $\sqrt{\#\text{parameters}}$, on Living-17 at pretrained initialization

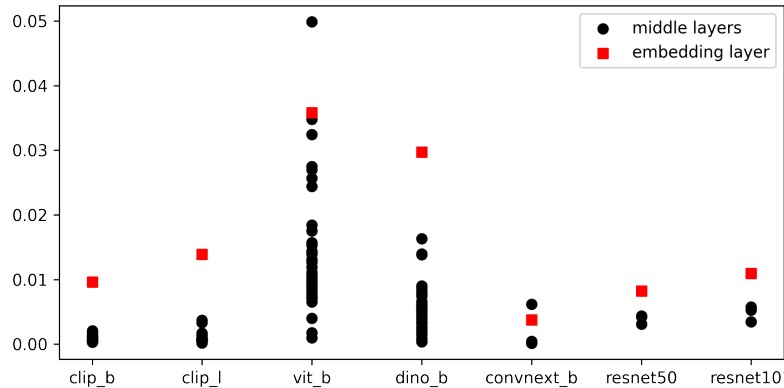

(c) Layer-wise gradient norms divided by $\sqrt{\#\text{parameters}}$, on Waterbirds at pretrained initialization

Figure 8: We visualize the layer-wise gradient norm, **divided by the square root of the number of parameters** on (a) DomainNet, (b) Living-17, and (c) Waterbirds, at the pretrained initialization. For better visualization, we omit the head from the plot which has predictably much larger than the others (since it is randomly initialized). The format is the same as Figure 2: gradient norms of "embedding" and "middle" layers are shown as **red-squares** and **black-circles**, respectively. Under this normalization, we see that the gradients of the embedding layer are much larger than the other layers in all models, including ResNets.

