# OpenReview forum: "How to Fine-Tune Vision Models with SGD"
_ICLR.cc/2024/Conference — ICLR 2024 poster_

### Official Review · Reviewer_qVNV · 2023-11-01

**Soundness:** 3 good
**Presentation:** 4 excellent
**Contribution:** 3 good
**Rating:** 6
**Confidence:** 4

**Summary:**

- The paper investigates the differences in utilizing AdamW vs. SGD for finetuning (not pretraining) modern ViT and Conv nets
- While usually AdamW outperforms SGD for finetuning, it finds that freezing the embedding layer and then applying SGD (with or without momentum) results in performance which is at par or better than AdamW on distribution shift benchmarks (WILDS-FMoW, WILDS-Camelyon, BREEDS-Living-17, Waterbirds, and DomainNet)
- The paper points out that models pretrained with AdamW do not finetune well with SGD, and that in such situations the embedding layer has higher grad norms than the rest of the parameters
- The paper produces SOTA results for the datasets when finetuning CLIP and ConvNeXt models with either SGD + freeze-embed or AdamW

**Strengths:**

- The paper looks into the well known phenomenon of SGD finetuning not matching up to AdamW for certain situations in vision models and finds some interesting observations
  - SGD generally only outperforms on the considered evaluations when pretraining is performed with AdamW, when SGD pretraining is used then there isn't much of a gap
  - The embedding layer has a very high grad norm when pretraining with AdamW which potentially results in SDG updating its weights too quickly, which is why the freezing operation improves SGD's results significantly
- The paper is well written and easy to read, and the experiments are convincing. The additional experiments in the appendix are also very useful to confirm the hypotheses presented in the paper.

**Weaknesses:**

- While the paper's focus is distribution shift evaluations, it doesn't touch upon conventional downstream evaluation tasks such as ImageNet-1k, iNaturalist, datasets in VTAB (https://github.com/google-research/task_adaptation), etc. Even if some of the conclusions hold only for these distribution shift datasets, it is still important to contrast the observations around grad norms and SGD vs. AdamW vs. SGD (freeze-embed) on other common visual transfer tasks.
- In Section 4 it is mentioned "SGD performs well when fine-tuning data is closer to pretraining." and the conclusion is drawn from just a single data point, CLIP, which is trained on very different data and very different losses. There isn't enough data to back this claim.
  - Minor: It also says that all models except CLIP were trained on ImageNet-21K, whereas DINO is trained on ImageNet-1k.
- The CIFAR-10 results on CLIP are presented in a misleading manner "SGD (freeze-embed) gets 20% and 30% lower error than SGD on CLIP ViT-B/16 and ViT-L/14, respectively." where the difference across SGD, AdamW and SGD (freeze-embed) is only <=0.4%
- The paper does a grid search on a single hyperparameter, learning rate, and sweeps over 6 values, but holds the weight decay constant at 0 for SGD and 0.01 for AdamW. Models pretrained differently (SSL, CLIP, supervised) usually require very different finetuning settings, so this search is a bit limited, but this isn't a major weakness since the primary point is showing that SGD (freeze-embed) catches up to AdamW without recipe changes to SGD.
- Minor: The paper claims contradictory numbers for SGD and AdamW memory consumption (12 bytes vs 16 bytes per parameter in the Abstract, and AdamW maintaining 2x to 3x more states as SGD in the Introduction).

**Questions:**

It is not clear (1) when researchers should use SGD (freeze-embed) as compared to SGD and AdamW and (2) in which situations the high grad norms for the embedding layer happen. Is this paper only relevant for tasks with a domain shift from pretraining, or for OOD evaluations? How does it work for more popular and / or in-domain transfer learning tasks, such as ImageNet or iNaturalist?

The paper does cite other works which can get similar performance with AdamW and SGD on ImageNet or iNaturalist, but it would still be worthwhile to show results with the hyperparameters in the paper and with SGD (freeze-embed).

A better understanding outside of the distribution shift datasets will help me bump up my rating.

---

> ### Author Response · Authors · 2023-11-23
> **(1/2) CIFAR-10 results, popular evaluation benchmarks, when to use SGD (freeze-embed), when high grad norms happen**
>
> We thank the reviewer for the thorough review and the positive comments. We address the reviewer's questions and concerns.
>
> > While the paper's focus is distribution shift evaluations, it doesn't touch upon conventional downstream evaluation tasks such as ImageNet-1k, iNaturalist, datasets in VTAB
> 1. We present results for **CIFAR-10** in Table 6.
> 2. We do not test on ImageNet because some of the models are pretrained using supervised labels on ImageNet.
> 3. We picked distribution shift datasets because we wanted to examine the accuracy **both in-distribution and out-of-distribution** (a critical problem in modern deep learning). We note that the datasets we use are **popular evaluation benchmarks**: as one metric, DomainNet has 1300 cites, WILDS has 900 cites, Waterbirds has 1100 cites, which is comparable to iNaturalist and VTAB. For some recent examples of transfer learning papers that focus on these datasets, see Kumar et al. 2022, Goyal et al., 2022, Ghosal et al., 2022, Wortsman et al. 2022. We also examine real world datasets in satellite data (FMoW) and tumor detection (Camelyon).
>
> That said, we agree that testing on more datasets will solidify our claims and will add this to the future work section.
>
> > It is not clear… when researchers should use SGD (freeze-embed) as compared to SGD and AdamW
> 1. **SGD-freeze embed works consistently well**, so we recommend using it as a general guideline. SGD freeze-embed does better than SGD in 80% of the 70 settings. In table 1 (out-of-distribution), we show 35 model-dataset pairs (7 models, and 5 datasets). SGD freeze-embed does better than SGD in 27/35 of the comparisons. In table 2 (in-distribution), SGD freeze-embed does better than SGD in 29/35 comparisons.
> 2. In modern deep learning it's rarely the case that a single optimizer dominates in every case—our paper gives a detailed analysis as to when and why each of the optimizers can do better. To get further improvements, practitioners can look at the **embedding layer gradients** at the start of training (which is very cheap). If these are similar to (or smaller than) the gradients of other layers, then we expect SGD (even without freeze embed) to work fairly well.
> 3. In addition, our experiments and analysis in Appendix E show that SGD works better when the model was pretrained with SGD, or when the fine-tuning data is similar to pretraining (e.g., Living-17, which is a subset of ImageNet). So researchers and practitioners can also use their understanding of the pretraining optimization process, and the relationship between the pretraining and fine-tuning data, to get further improvements.
>
> > It is not clear… in which situations the high grad norms for the embedding layer happen
>
> We examine this question in detail in Appendix E. High grad norms for the embedding layer happen when the model is pretrained with AdamW. Rough intuition: AdamW normalizes the gradient by the variance—so there is "no incentive" to equalize the gradient norms across different layers.

---

> ### Author Response · Authors · 2023-11-23
> **(2/2) we did try weight decay, writing, and analysis**
>
> > The paper does a grid search on a single hyperparameter… holds the weight decay constant
>
> We **did try SGD with weight decay** of 0.01 (see tables 7 and 8 in the Appendix) but this did not fix the issue with SGD. The average OOD accuracy on a CLIP ViT-B/16 was 67.8\% for SGD with weight decay, 67.9\% for SGD without weight decay, and 75.9\% for SGD (freeze-embed). We agree that we could further tune the weight decay for AdamW and SGD, but it was already expensive to sweep across learning rates $\times$ datasets $\times$ models $\times$ optimizers.
>
> > The CIFAR-10 results on CLIP are presented in a misleading manner "SGD (freeze-embed) gets 20% and 30% lower error than SGD on CLIP ViT-B/16 and ViT-L/14, respectively." where the difference across SGD, AdamW and SGD (freeze-embed) is only <=0.4%
>
> Just to clarify, for a ViT-L, SGD got **99.0%** accuracy, and AdamW and SGD (freeze-embed) got **99.3%**. We calculated **error reduction** as (99.3% - 99.0%) / (100% - 99.0%) = **30%**. The reason we examine the error reduction is that **at such high accuracy levels**, it is much more **difficult to improve the accuracy**. For example, it is often not too hard to improve the accuracy from 80.0% to 82.0% (a 2% improvement), but improving the accuracy from 99.0% to 101% is impossible. In other words the 0.3% accuracy improvement should be contextualized with the maximum possible accuracy improvement of 1.0%. However, we acknowledge your point and have **edited to note the exact accuracy** in the text. Happy to make additional changes as you like.
>
> > In Section 4 it is mentioned "SGD performs well when fine-tuning data is closer to pretraining." and the conclusion is drawn from just a single data point, CLIP
>
> Just to clarify—our claim in Section 4 is not based on CLIP. We examined all the models pretrained on ImageNet (multiple models), and compared their performance across multiple datasets. We noticed that SGD performs well on the dataset that is most similar to ImageNet (Living-17, which is a subset of ImageNet). We agree that more evidence would be good—we have softened the phrasing to "This suggests that SGD **may** work better when fine-tuning and pretraining data are similar—we leave a more thorough analysis to future work". Happy to make more changes—how does that sound?
>
> > Minor: It also says that all models except CLIP were trained on ImageNet-21K, whereas DINO is trained on ImageNet-1k.
>
> Great catch, we have fixed this.
>
> > "contradictory numbers for SGD and AdamW memory consumption (12 bytes vs 16 bytes per parameter in the Abstract, and AdamW maintaining 2x to 3x more states as SGD in the Introduction)."
>
> Thank you for pointing this out. We agree and apologize for the mistake. To clarify:
> 1. SGD (no momentum) stores 2 floats per parameter
> 2. SGD (with momentum) stores 3 floats per parameter
> 3. AdamW stores 4 floats per parameter
>
> We have fixed the text to say that AdamW maintains 1.33$\times$ to 2$\times$ more states as SGD in the introduction.\*
>
> \* Calculation: 4 / 3 = 1.33 and 4 / 2 = 2

---

### Official Review · Reviewer_zcmx · 2023-11-01

**Soundness:** 3 good
**Presentation:** 3 good
**Contribution:** 3 good
**Rating:** 6
**Confidence:** 4

**Summary:**

This paper unveils a captivating strategy for enhancing the fine-tuning of large vision models (e.g., Vision Transformer, ConvNeXt models, etc.). While SGD and AdamW are both widely used, SGD is favored for its memory efficiency when they perform equally well. However, the authors reveal that especially in scenarios with *distribution shifts*, AdamW excels in fine-tuning modern Vision Transformer and ConvNeXt models.

The key observation is that this performance gap between SGD and AdamW is most pronounced when the gradients in the first "embedding" layer are significantly larger than those in the rest of the model. To address this issue, the authors propose a simple solution: freezing the embedding layer, which accounts for less than 1% of the parameters. This approach not only matches AdamW's performance but also conserves memory, with up to a 33% reduction in GPU usage on ViT-L.

The proposed approach leads to SOTA results on five popular distribution shift benchmarks, such as WILDS-FMoW, WILDS-Camelyon, BREEDS-Living-17, Waterbirds, and DomainNet. In essence, this paper offers an enchanting recipe to narrow the performance gap between SGD with momentum and AdamW. Their findings shed light on optimizing large vision models.

**Strengths:**

1. This paper is well-written, the motivation is sound, and the problem has been addressed well.
2. The paper addresses the crucial issue of improving out-of-distribution (OOD) accuracy when fine-tuning vision models.
3. The paper achieves state-of-the-art results on multiple benchmarks, demonstrating its efficacy in narrowing the performance gap between SGD and AdamW.

**Weaknesses:**

1. Although the experimental results achieve SOTA performance and practical memory savings, the improvement of the proposed approach, compared to the vanilla approach is marginal.

2. The experimental results show that using SGD (freeze-embed) may also result in poor performance in some cases, how does the author explain this phenomenon?

3. Reprogramming [1, 2] and Visual Prompting [3, 4, 5] is also a parameter-efficient finetuning approach, please discuss these methods in the related work.

> [1] Elsayed et al. Adversarial Reprogramming of Neural Networks (ICLR 2019)
>
> [2] Chen. Resource-Efficient Cross-Domain Machine Learning
>
> [3] Bahng et al. Exploring Visual Prompts for Adapting Large-Scale Models
>
> [4] Tsao et al. AutoVP: An Automated Visual Prompting Framework and Benchmark
>
> [5] Jia et al. Visual Prompt Tuning (ECCV 2022)

**Questions:**

Please refer to the weakness. In addition, please address the following questions:

1. It is clear that the proposed approach is not always the best choice, how does the developer or engineer decide which approach to use when it comes to different datasets?

2.  Please report the mean and standard deviation of the results across multiple rounds (different seeds).

---

> ### Author Response · Authors · 2023-11-23
> **Response: improvement of proposed approach, which method to use, more cites, mean and std-dev reported for CLIP**
>
> We thank the reviewer for the useful review, and for saying that our paper "addresses the crucial issue", "the problem has been addressed well", and for recognizing the "state-of-the-art results". We address the reviewer's questions below.
>
> > the improvement of the proposed approach, compared to the vanilla approach is marginal
>
> We note that the improvements of SGD (freeze embed) are quite significant:
> - Comparison to SGD: SGD (freeze-embed) achieves **5% higher accuracy OOD**, averaged across all datasets and models, and also performs better on the vast majority of datasets.
> - Comparison to AdamW: SGD (freeze-embed) uses substantially less memory. For example, **AdamW consumes 50% more memory** than SGD (freeze-embed, no momentum) on a ViT-L---on top of the significant memory savings, SGD (freeze-embed, no momentum) still achieves higher OOD accuracy.
>
> > how does the developer or engineer decide which approach to use when it comes to different datasets?
>
> 1. **SGD-freeze embed works consistently well**, so we recommend using it as a general guideline. SGD freeze-embed does better than SGD in 80% of the 70 settings (7 models $\times$ 5 datasets $\times$ {ID, OOD}).  In table 1 (out-of-distribution), we show 35 model-dataset pairs (7 models, and 5 datasets). SGD freeze-embed does better than SGD in 27/35 of the comparisons. In table 2 (in-distribution), SGD freeze-embed does better than SGD in 29/35 comparisons.
> 2. In modern deep learning it's rarely the case that a single optimizer dominates in every case—our paper gives a detailed analysis as to when and why each of the optimizers can do better. To get further improvements, practitioners can look at the **embedding layer gradients** at the start of training (which is very cheap---a single backward pass). If these are similar to (or smaller than) the gradients of other layers, then we expect SGD (even without freeze embed) to work fairly well.
> 3. In addition, our experiments and analysis in Appendix E show that SGD works better when the model was pretrained with SGD, or when the fine-tuning data is similar to pretraining (e.g., Living-17, which is a subset of ImageNet). So researchers and practitioners can also use their understanding of the pretraining optimization process, and the relationship between the pretraining and fine-tuning data, to get further improvements.
>
> > Reprogramming [1, 2] and Visual Prompting [3, 4, 5]... please discuss these methods in the related work.
>
> Thank you for the useful pointers—we have **added these to the related works**.
>
> > Please report the mean and standard deviation of the results across multiple rounds (different seeds).
>
> We **reported mean and std-dev for CLIP** (the most advanced of the models we consider for distribution shift) across multiple seeds in Table 5.
> Running multiple seeds for all of table 2 was too expensive—we already did a sweep over 4 optimizers x 5 datasets x 7 models x 6 learning rates (and also other optimizers such as LARS and LAMB, ablations on weight decay, etc)—multiplying all results by multiple seeds was computationally infeasible. However, we hope that the multiple seeds for CLIP, and breadth of results across datasets and models, is helpful. If there are any specific results you are particularly interested in, we can run additional seeds for them!

---

> ### Comment · Reviewer_zcmx · 2023-11-27
>
> Thanks a lot for the authors' response. I believe the authors' revisions to the paper have further improved their work. I am inclined to recommend accepting this paper.

---

### Official Review · Reviewer_6FtM · 2023-11-04

**Soundness:** 3 good
**Presentation:** 3 good
**Contribution:** 3 good
**Rating:** 6
**Confidence:** 4

**Summary:**

The paper proposes a method to fine-tune pre-trained models using SGD instead of AdamW to reduce the amount of memory the optimizer states take. The main observation of the paper is that when fine-tuning on out of distribution data the gradient magnitudes on the first embedding layer for architectures pre-trained with AdamW and fine-tuned with SGD are large. The authors hypothesize that the large gradients result in overfitting on OOD data. To address the paper freezes the embedding layer when fine-tuning with SGD and shows that the models can be fine-tuned as well as or better than using AdamW with lower memory footprint. The experimental evaluation and the results validate the hypothesis and the effectiveness of the proposed freeze SGD fine-tuning method.

**Strengths:**

* As the models are become larger reducing the memory footprint to train/fine-tune models becomes increasingly important. One way to reduce the memory footprint is to use optimizers which do not use additional much state. The paper shows that by freezing a the initial embedding layer SGD with momentum or just plain SGD can match performance of fine-tuning the full model with AdamW.
* The ablation studies show interesting observations on the role of the optimizer difference when pre-training and fine-tuning. Some of these observations can help practitioners in choosing the optimizer and hyper parameters when fine-tuning.

**Weaknesses:**

* All the down stream fine-tuning experiments are focused on classification. It would be good to see if the observations hold for other tasks like detection, segmentation etc.
* Gradual unfreezing is mentioned in Table 5 but the method is not clearly described in the paper. There is some mention of the learning rate schedule with gradual unfreezing in the appendix but does not fully explain what was done.

**Questions:**

* Given an optimizer mismatch can the overfitting of early layers be avoided by low learning rates in the early part of fine-tuning? Overall the learning rate schedule seems to have an important role to play when there is an optimizer mismatch. Have the authors tried using different learning rate schedules and are the large gradient problems persistent throughout the optimization or confined to initial steps when fine-tuning.

---

> ### Author Response · Authors · 2023-11-23
> **Response on more applications, gradual unfreezing, learning rate schedules**
>
> We thank the reviewer for the positive and useful review, and for expressing that the soundness, contribution, and presentation of the paper are good. We address the remaining questions and concerns below.
>
> > good to see if the observations hold for other tasks like detection, segmentation etc.
>
> This is a good idea! We wanted to be thorough and prioritize covering different types of datasets, distributions shifts, and models for this work, but examining segmentation and detection is a good idea for future work. We note that follow-up work has shown that SGD (freeze-embed) works well for NLP models such as RoBERTa—so it is more general than image classification.
>
> > Gradual unfreezing is mentioned in Table 5 but the method is not clearly described in the paper.
>
> We apologize for the confusion. Gradual unfreezing is another ablation we tried—it is not a main contender in the paper, but we give results in Appendix B. The idea is to unfreeze layers from top to bottom over the course of training. We have added a description to Appendix B.
>
> > Given an optimizer mismatch can the overfitting of early layers be avoided by low learning rates in the early part of fine-tuning?
>
> This is an interesting idea! We note that learning rate warmup is generally helpful for AdamW (because it leads to better estimates for the moments), and generally has a smaller benefit for SGD. Also, we did sweep over many learning rates including some very small learning rates and selected the best, which hopefully partially accounts for this. In this work we focused our additional experiments on different optimizers (e.g., AdamW, LARS, LAMB), lower learning rates on the embedding layer, linear probing, SGD with weight decay, etc.
>
> We agree that the learning rate schedule is good to explore in future work, and have mentioned this in the revised submission.

---

### Official Review · Reviewer_RNj8 · 2023-11-04

**Soundness:** 3 good
**Presentation:** 3 good
**Contribution:** 3 good
**Rating:** 6
**Confidence:** 3

**Summary:**

This paper found that Adam achieves better OOD and ID accuracy than SGD, mainly because the embedding layer of some currently popular models yields larger gradients. Such larger gradients can be well treated by Adam, but may cause over-training of the embedding layer by SGD. Therefore, this work proposes two variants of SGD (freezing the embedding layer; freezing the embedding layer and no momentum), which are found to boost the accuracy significantly.

**Strengths:**

1. Very extensive experiments are conducted and thorough analysis is provided.
2. The presentation is good and motivation is clear and strong.
3. The minor yet effective modification on SGD shows good performance improvement, with less memory requirement than Adam.

**Weaknesses:**

Since I am not researching around the optimization domain, I can not clearly point out what is the weakness of this paper. See my questions below.

**Questions:**

1. Will such modified SGD still achieve performance gain on NLP model such as Llama2?
2. As you mentioned, the reason why the early embedding layer has larger gradients is that such models are typically pre-trained by Adam. What will the result be if they are models are pre-trained by SGD? Will it be worse than with Adam?

---

> ### Author Response · Authors · 2023-11-23
> **Responses: NLP Models, pretrained with SGD**
>
> We thank the reviewer for the positive and useful review, and for recognizing the "very extensive experiments", "thorough analysis", and "good performance improvement". We address the remaining questions:
>
> > Will such modified SGD still achieve performance gain on NLP model such as Llama2?
>
> We note that follow-up work has shown that SGD (freeze-embed) **works well for NLP models** such as RoBERTa. We think examining whether it helps Llama2 is a great avenue for future work—based on our results in Appendix E we believe this is potentially promising because Llama2 was pretrained using AdamW, and that is when our method tends to outperform SGD.
>
> > As you mentioned, the reason why the early embedding layer has larger gradients is that such models are typically pre-trained by Adam. What will the result be if they are models are pre-trained by SGD? Will it be worse than with Adam?
>
> In Appendix E, we run controlled experiments where we pretrain a ResNet-50 model with (a) SGD, or (b) AdamW. We then fine-tune each of these models with (1) SGD, (2) AdamW, and (3) SGD Freeze-Embed. We find that **when the model is pretrained with SGD, then it is better to fine-tune with SGD than with AdamW.**

---

> ### Author Response · Authors · 2023-11-23
> **Change in rating**
>
> Dear reviewer, we noticed that your rating was changed from 8 to 6 recently---we just wanted to check in if you had any follow-up thoughts or concerns that we can answer! Unfortunately due to major personal issues, I was not able to respond to the review until the end of the rebuttal period. Sorry about that. We really appreciate your time and comments, and hope these answers address your concerns. We would love to address any remaining concerns! Thank you!

---

### Official Review · Reviewer_S1Jb · 2023-11-06

**Soundness:** 4 excellent
**Presentation:** 4 excellent
**Contribution:** 3 good
**Rating:** 8
**Confidence:** 3

**Summary:**

This paper analyzes why AdamW often outperforms SGD when finetuning large pretrained vision models. It turns out that the embedding layer gradients are quite large and AdamW is able to suppress the huge swings in embedding layer values caused by these gradients. A simple method is proposed whereby the embedding layer is frozen during finetuning. This method is shown to perform on par or better than AdamW in both in-distribution and out of distribution finetuning. Moreover, SGD with frozen embedding layer uses less memory than AdamW (roughly 1.3x less).

**Strengths:**

I found the paper to be extremely well written. All claims were backed up by experimental results. The method is simple, yet effective. Very good paper.

**Weaknesses:**

1) It seems that the entire paper hinges on models which have an embedding layer. While these models are popular now, they may not be popular forever, which limits the long term impact of this method.
2) In models beyond vision, such as recommendation models (i.e. DLRM https://arxiv.org/abs/1906.00091), the embedding layer contains most of the model parameters. In such cases, it is also unclear whether this kind of method could work. Of course, the authors are explicit that this paper is about vision models in particular, so I am not penalizing them for this (possible) shortcoming
3) In some cases, SGD performed as well or better than the other variants. From the practitioners point of view, how should one choose which method to use?

**Questions:**

1) Why does the memory overhead of adamw relative to sgd increase with model size? It seems like the relative difference in memory overhead should be constant
2) For Fig. 2, are the gradient norms normalized by number of elements?
3) Why do you think SGD outperforms the other alternatives across so many experiments?

---

> ### Author Response · Authors · 2023-11-23
> **Thank you for positive review; responses on choosing methods, memory overhead, and more**
>
> We thank the reviewer for expressing that this is a "very good paper", and for the helpful comments. We respond to their questions below.
>
> > From the practitioners point of view, how should one choose which method to use?
>
> 1. **SGD-freeze embed works consistently well**, so we recommend using it as a general guideline. SGD freeze-embed does better than SGD in 80% of the 70 settings (7 models $\times$ 5 datasets $\times$ {ID, OOD}).
> 2. In modern deep learning it's rarely the case that a single optimizer dominates in every case—our paper gives a detailed analysis as to when and why each of the optimizers can do better. To get further improvements, practitioners can look at the **embedding layer gradients** at the start of training (which is computationally very cheap---a single backward pass). If these are similar to (or smaller than) the gradients of other layers, then we expect SGD (even without freeze embed) to work fairly well.
> 3. In addition, our experiments and analysis in Appendix E show that SGD works better when the model was pretrained with SGD, or when the fine-tuning data is similar to pretraining (e.g., Living-17, which is a subset of ImageNet). So researchers and practitioners can also use their understanding of the pretraining optimization process, and the relationship between the pretraining and fine-tuning data, to get further improvements.
>
> > Why does the memory overhead of adamw relative to sgd increase with model size?
>
> This is because there is **memory** taken up by **activations (same for all optimizers)**, and by **parameters and optimizer state** (which depends on the optimizer).
> - Memory of AdamW = Activation memory + 4 $\times$ parameter memory
> - Memory of SGD (no momentum) = Activation memory + 2 $\times$ parameter memory
>
> For **larger ViT models, parameter memory dominates activation memory**, and so the overhead of AdamW increases (up to a maximum of 2x).
>
> > hinges on models which have an embedding layer
> > recommendation models… embedding layer contains most of the model parameters
>
> Some recent follow-ups to our paper have found that freezing the embedding layer can lead to improvements in NLP models such as RoBERTa.
>
> Conceptually, we can think of the embedding layer as an abstract interface between the raw input and the later networks in the model, in which case our kind of analysis may be useful for a more general class of models.
>
> Recommender systems is another interesting domain—we have noted this and the cite in the future work section.
>
>
> > For Fig. 2, are the gradient norms normalized by number of elements?
>
> Figure 2 does not normalize by number of elements, however in Appendix F we show figures where they are normalized by the parameter norm, and the number of parameters.
>
> > Why do you think SGD outperforms the other alternatives across so many experiments?
>
> Could we please clarify which alternative you are referring to here? SGD (freeze-embed) and AdamW generally perform better than SGD in the majority of experiments

---

### Meta-Review · Area_Chair_HkkM · 2023-12-08

**Metareview:**

- The paper explores fine-tuning pre-trained models, specifically contrasting SGD with AdamW. It observes that fine-tuning with SGD on out-of-distribution data can cause overfitting due to large gradient magnitudes in the first embedding layer, a challenge not faced with AdamW. To mitigate this, the paper proposes an SGD variant: freezing the embedding layer. The approach, especially effective in models pre-trained with AdamW, shows comparable or superior performance to AdamW on distribution shift benchmarks like WILDS-FMoW and BREEDS-Living-17.

**Justification For Why Not Higher Score:**

- As also pointed out by Reviewer qVNV, It is strange that the submission does not report the results in the most common setting on ImageNet-1K.
- The method can only be applied to the fine-tuning scenario while the memory cost in the pre-training is still important.

**Justification For Why Not Lower Score:**

- The current result seems impressive for OOD cases.
- All the reviewers agree that the advantages of this submission outweigh its disadvantages.

---

### Decision · Program_Chairs · 2024-01-16

Accept (poster)